# The influence of tides on the marine carbonate chemistry of a coastal polynya in the south-eastern Weddell Sea

Elise S. Droste[1], Mario Hoppema[2], Melchor González-Dávila[3], Juana Magdalena Santana-Casiano[3], Bastien Y. Queste[1,4], Giorgio Dall'Olmo[5,10], Hugh J. Venables[6], Gerd Rohardt[2], Sharyn Ossebaar[7], Daniel Schuller[8], Sunke Trace-Kleeberg[2,9], and Dorothee C. E. Bakker[1]

[1]School of Environmental Sciences, University of East Anglia, Norwich Research Park, NR4 7TJ Norwich, United Kingdom
[2]Alfred Wegener Institute Helmholtz Centre for Polar and Marine Research, Postfach 120161, 27515 Bremerhaven, Germany
[3]Instituto de Oceanografía y Cambio Global, IOCAG, Universidad de Las Palmas de Gran Canaria, ULPGC, 35017 Las Palmas de Gran Canaria, Spain
[4]Department of Marine Sciences, University of Gothenburg, Carl Skottsbergs Gata 22B, SE-413 19 Gothenburg, Sweden
[5]Plymouth Marine Laboratory, Prospect Place, PL1 3DH Plymouth, United Kingdom
[6]British Antarctic Survey, High Cross, Madingley Road, CB3 0ET Cambridge, United Kingdom
[7]Royal Netherlands Institute for Sea Research (NIOZ), Department of Ocean Systems, P.O. Box 59, 1790 AB, Den Burg, Texel, The Netherlands
[8]Scripps Institution of Oceanography, UC San Diego, 8622 Kennel Way, La Jolla, CA 92037, United States
[9]School of Ocean and Earth Science, National Oceanography Centre, University of Southampton, SO14 3ZH Southampton, United Kingdom
[10]Istituto Nazionale di Oceanografia e di Geofisica Sperimentale, Borgo Grotta Gigante 42/c, 34010 Sgonico, Trieste, Italy

**Correspondence:** Elise Droste (e.droste@uea.ac.uk)

**Abstract.**

Tides significantly affect polar coastlines by modulating ice shelf melt and modifying shelf water properties through transport and mixing. However, the effect of tides on the marine carbonate chemistry in such regions, especially around Antarctica, remains largely unexplored. We address this topic with two case studies in a coastal polynya in the south-eastern Weddell Sea, neighbouring the Ekström Ice Shelf. The case studies were conducted in January 2015 (PS89) and January 2019 (PS117), capturing semi-diurnal oscillations in the water column. These are pronounced in both physical and biogeochemical variables for PS89. During rising tide, advection of sea ice meltwater from the north-east created a fresher, warmer, more deeply mixed water column with lower dissolved inorganic carbon (DIC) and total alkalinity (TA) content. During ebbing tide, water from underneath the ice shelf decreased the polynya's temperature, increased the DIC and TA content, and created a more stratified water column. The variability during the PS117 case study was much smaller, as it had less sea ice meltwater input during rising tide and was better mixed with sub-ice shelf water. The contrasts in the variability between the two case studies could be wind and sea ice driven, and underline the complexity and highly dynamic nature of the system.

The variability in the polynya induced by the tides results in an air-sea $CO_2$ flux that can range between a strong sink (-24 mmol m$^{-2}$ day$^{-1}$) and a small source (3 mmol m$^{-2}$ day$^{-1}$) on a semi-diurnal time scale. If the variability induced by tides is not taken into account, there is a potential risk of overestimating the polynya's $CO_2$ uptake by 67 % or underestimating it by 73 %, compared to the average flux determined over several days. Depending on the timing of limited sampling, the polynya may appear to be a source or a sink of $CO_2$. Given the disproportionate influence of polynyas on heat and carbon exchange in

polar oceans, we recommend future studies around the Antarctic and Arctic coastlines to consider the timing of tidal currents in their sampling strategies and analyses. This will help constrain variability in oceanographic measurements and avoid potential

biases in our understanding of these highly complex systems.

## 1 Introduction

Tides in polar regions have recently gained increasing attention in investigations focusing on understanding the physical interaction between the ocean and sea ice (e.g., Dmitrenko et al., 2012; Kirillov et al., 2013; Skogseth et al., 2013) and how

they affect ice shelf melt (e.g., Padman et al., 2009; Makinson et al., 2011; Mueller et al., 2018; Padman et al., 2018; Huot et al., 2021). Tidal effects on the biogeochemistry along polar coastlines, however, have not yet received similar attention, even though tides have been shown to affect the chemical properties and the fugacity of seawater $CO_2$ ($f CO_2$) in other regions (Rogachev et al., 2001; Andersson and MacKenzie, 2012; Sims et al., 2022).

The tides in the Weddell Sea are the largest in the Southern Ocean (Padman et al., 2018). Here, relatively warm, carbon-rich

Warm Deep Water (WDW) upwells in the east and is physically and chemically altered along its route towards the western Weddell Sea, a hotspot for Antarctic Bottom Water (AABW) formation and carbon sequestration (Fahrbach et al., 1994; Anderson et al., 1991; Huhn et al., 2013). Water on the eastern shelf also has the potential to be exported to the deep ocean by northward transport into the Upper Circumpolar Deep Water (uCDW) (Orsi et al., 2002), along with any modifications to the dissolved inorganic carbon (DIC) content on the continental shelf. Studying processes that modify the physical and chemical

properties of these water masses on the shelf, such as tides, will enable a better understanding of the transport of carbon to the deep ocean.

A common feature along the Antarctic coastline are coastal polynyas, described as areas of open water where sea ice cover is expected. Absence of sea ice enables strong direct interaction between the ocean and the atmosphere in regions and at times of the year when sea ice cover would otherwise restrict it and strong gradients between ocean and atmosphere exist. Polynyas

are therefore thought to have a disproportionally large impact on the polar oceans considering their relatively small surface area (Barber and Massom, 2007). The direct exposure of the ocean to the atmosphere has an impact on the marine carbonate system, altering the seawater $f CO_2$ and thus the capacity for ocean $CO_2$ uptake. Whether the polynya is a source or a sink of $CO_2$ strongly depends on the interplay between processes that increase the seawater $f CO_2$ beyond the atmospheric $f CO_2$, and those that decrease it. For example, polynyas can have an enhancing role in sea ice production as they can release heat

directly to the atmosphere (Renfrew, 2002). In this process, brine, including DIC, is rejected from the sea ice matrix into the ocean (Rysgaard et al., 2011; Skogseth et al., 2013). In the summer, dilution by sea ice melt and calcium carbonate ($CaCO_3$) dissolution draws down $CO_2$ (Rysgaard et al., 2011). Additionally, enhanced light availability supports biological productivity (Arrigo and van Dijken, 2003), which consumes $CO_2$ that can thereby typically generate a strong $CO_2$ sink (Yager et al.,

1995; Arrigo et al., 2008). The prolonged ice-free conditions also potentially allow enough time for the polynya to take up
as much $CO_2$ as is necessary to reach equilibrium with the atmosphere (Hoppema and Anderson, 2007). However, biological
productivity is found to be variable among Antarctic coastal polynyas (Arrigo et al., 2015), and in some cases is not sufficient
to keep a polynya from outgassing $CO_2$ (Arroyo et al., 2019). Tides can displace water on short timescales (Skogseth et al.,
2013; Llanillo et al., 2019) and modify the water column structure through enhanced mixing (Padman et al., 2009). On the
one hand, tidal mixing in coastal polynyas can replenish nutrients at the surface (Tremblay et al., 2002), stimulate biological
production, and thereby enhance $CO_2$ drawdown. On the other hand, mixing with $CO_2$-rich deep water can increase surface
water $f CO_2$ (Rogachev et al., 2001), or in some cases erode a stable mixed layer necessary to support phytoplankton growth
(Arrigo and van Dijken, 2003).

The effects of tides on the marine carbonate chemistry of these biogeochemically impactful regions along the Antarctic
coastline remain largely unexplored. Understanding the tidal influence may help us to quantify some of the variability observed
among Antarctic polynyas, such as the variability in the biological productivity (Arrigo et al., 2015) as well as the capacity of
coastal polynyas to absorb or release atmospheric $CO_2$ (Arroyo et al., 2019). Practically, this improved understanding can help
develop more reliable sampling activities that consider the timing and strength of tidal currents, thereby obtaining representative
data of a highly dynamic system.

In this work, we illustrate the effect of tides on the marine carbonate system with two case studies in a coastal polynya in the
south-eastern Weddell Sea (Fig. 1). We present biogeochemical observations for two tidal sampling campaigns in the austral
summer. To the best of our knowledge, this is the first time that tidal influences on the seawater carbonate chemistry are studied
in a coastal Antarctic polynya. We discuss the variability induced by tidal forces and explore the differences between the two
tidal observations in terms of their physical and biogeochemical characteristics, and what it means for ocean $CO_2$ uptake.

## 2   Methods

### 2.1   Sampling location

The data were collected during two repeat hydrographic expeditions in the Weddell Sea with the German icebreaker *R.V.*
*Polarstern*: expedition PS89 (2 December 2014 - 31 January 2015; Cape Town - Cape Town; (Boebel, 2015)) and expedition
PS117 (12 December 2018 - 7 February 2019; Cape Town - Punta Arenas; (Boebel, 2019)). During both expeditions, the tidal
cycle was recorded by means of repeat CTD casts at the same location at a frequency that was high enough to constrain the
tidal oscillation in the water column. These recordings will hereafter be referred to as *tidal observations*. The tidal observations
were performed at the same site in a coastal polynya, 56 km west of Atka Seaport, directly at the edge of the Ekström Ice
Shelf (Fig. 1), the geometry of which is thought to be representative of most ice shelves of Dronning Maud Land (Smith et al.,
2020a). The polynya is bordered by the Ekström Ice Shelf along its southern edge, and by sea ice along the rest of its perimeter.
It regularly forms in the summer months (Boebel, 2019; Arrigo and van Dijken, 2003; Arrigo et al., 2015). Coastal polynyas
in this region are typically formed by katabatic winds from the continent that advect the newly produced sea ice away from the

coastline in the spring and summer (Eicken and Lange, 1989; Renfrew, 2002). At the Ekström Ice Shelf, a number of grounded ice bergs to the east can also create a sea-ice free lee downstream from the westward-flowing coastal current (Boebel, 2019).

The size and shape of the polynyas at the Ekström Ice Shelf are highly variable, depending on the wind direction and speed. For the PS89 tidal observations, the average width (between the ice shelf and sea ice) of the Ekström polynya is about 0.8 km, estimated from Synthetic Aperture Radar (SAR) images (Fig. A1). The length was approximately 12 km. During the PS117 tidal observation, the polynya was substantially larger, having an approximate average width of 3 km and a length of about 40 km (Fig. A3). Given that the average area of coastal polynyas in the eastern Weddell Sea in summer is 7.75 x $10^3$ km$^2$ (coastal polynyas numbers 12-17 in Arrigo and van Dijken (2003)), the polynya during both tidal observations at this location is considered to be relatively small.

Information about the tidal observations is shown in Table 1. Sources of supporting data used in this study can be found in the Supplementary Materials (Table B1). Both tidal observations were made at the same time of year (i.e. mid-January). The sampling during PS89 (75 hours) lasted ∼3 times longer than during PS117 (22 hours). The exact location of the hydrographic stations differed slightly between the two expeditions due to shifting of the Ekström Ice Shelf extent between 2015 and 2019. The sampling sites for PS117 were chosen in such a way that the distance to the ice shelf was approximately the same as during PS89. Due to other ongoing scientific activities on board during the PS117 tidal observation, the exact location of sampling within the coastal polynya varied slightly per CTD cast, but the casts remained within a maximum distance of 300 m from each other.

## 2.2 Hydrographic and biogeochemical observations

In addition to a SBE911plus CTD sensor, each rosette sampler (SBE32, 24 x 12 L bottles) was equipped with a fluorometer (EcoFLR, Wetlabs), an oxygen optode (SBE43, Seabird Electronics), and an Acoustic Doppler Current Profiler (ADCP) system (2x 300 kHz RDI Workhorse ADCPs) (Table C1). The oxygen measurements were calibrated using discrete oxygen samples from deep CTD casts only, analysed with the Winkler method (Boebel, 2015, 2019; Rohardt and Tippenhauer, 2020). During the PS117 tidal observation period, two different CTD rosettes with their own set of sensors were used in alternation. On one of these two CTD rosettes, the oxygen optode sensor malfunctioned and thus its data had to be excluded from further analysis. The fluorescence data for this particular rosette is also not available. The minor implication of this is addressed in Section 3.3.

Dissolved inorganic carbon (DIC) and total alkalinity (TA) samples were collected following Dickson et al. (2007). DIC and TA samples were analysed on a VINDTA 3C system (Mintrop, 2016), which uses coulometry for DIC and potentiometric titration for TA determination. TA was calculated according to Dickson et al. (2007) for PS89 and using the Python package Calkulate (Humphreys and Matthews, 2022) for PS117. On PS89, dissolved phosphate ($PO_4^{3-}$), silicate ($SiO_4$), and nitrate ($NO_3^-$) analyses performed by UV-Vis spectrophotometric methods were carried out on board with a SEAL Analytical continuous-flow AutoAnalyzer (Boebel, 2015). During PS117, nutrient samples were analysed simultaneously for $PO_4^{3-}$, $SiO_4$, nitrite ($NO_2^-$), and the combination of $NO^{3-} + NO_2^-$ on board on a continuous gas-segmented flow TRAACS 800 Auto-Analyzer (Technicon, a.k.a. SEAL Analytical) within 4-5 hours after sampling (Boebel, 2019). Samples were calibrated

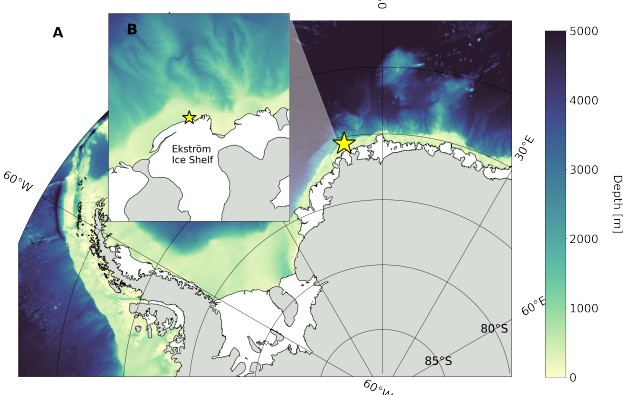

**Figure 1.** Map of study site: A) Weddell Sea with bathymetry: IBSCO Version 1.0 (Arndt et al., 2013). Grey regions represent the Antarctic continent and land-fast ice, white regions represent ice shelves. B) Enlarged map of coastal study site along the Ekström Ice Shelf. Yellow star shows location of tidal measurements. See Table 1 for coordinates.

with standards diluted in low nutrient seawater within the salinity range of the Southern Ocean. Analytical uncertainties can be found in Tables C2 and C3 .

To complement the ADCP data, a tidal model (Model CATS2008) was used to determine the times of high (rising) and low (ebbing) tide during the tidal experiments. This model has been optimised for the Antarctic seas, using available measurements and including ice shelf cavities to improve performance (Padman et al., 2002). All tidal constituents, which represent each mode of the tidal oscillation on a different time-scale, are used (e.g. M2, S2, K1). The modelled currents are averaged over the entire water column for tidal currents only, i.e. they are not total currents. Total currents (measured by the ADCP) include contributions from, for example, mean flow along the continental shelf.

## 2.3 The marine carbonate system and CO$_2$ flux calculations

The air-sea CO$_2$ flux was calculated according to:

$$F = kK_0(fCO_{2sw} - fCO_{2atm}) \tag{1}$$

$F$ is the CO$_2$ flux in mol m$^{-2}$ hr$^{-1}$, $k$ is the gas transfer velocity, $K_0$ is the CO$_2$ solubility at *in situ* temperature and salinity, and $fCO_2$ is the fugacity of CO$_2$ in seawater ($sw$) and in the atmosphere ($atm$). The gas transfer velocity is calculated according to the parameterisation of Wanninkhof (2014). Minimum, maximum, and average wind speed for the duration of the tidal observation period, as measured on board and reported at 10 m above sea level, were used as input to this parameterisation (Table 1). For the Schmidt number, i.e. the parameter that relates the gas transfer velocity of different gases, we use Wanninkhof (2014), which gives a refitted polynomial updated from Wanninkhof (1992) to cover a temperature range of -2 to 40 °C and is virtually the same as the parameterisation by Ho et al. (2006). The CO$_2$ solubility is calculated according to Weiss (1974). The

**Table 1.** Details of the tidal observations based on repeat CTD casts and discrete seawater sampling during the two hydrographic expeditions: PS89 and PS117. For both observation periods, casts were lowered into the water during times of ebbing and rising tide. Depth of the polynya was around 165 m and 200 m during PS89 and PS117, respectively.

|  | PS89 | PS117 |
| --- | --- | --- |
| Start observations [UTC] | 21:09 9 Jan. 2015[a] | 14:27 11 Jan. 2019 |
| End observations [UTC] | 01:00 11 Jan. 2015 | 11:57 12 Jan. 2019 |
| Location latitude | 70° 31′ 24″ S | 70°31′ 19.56″ S |
| Location longitude | 8° 45′34.2″ W | 8°46′ 6.76″ W |
| Deepest depth of CTD cast [m] | 160 | 190.5 - 201.5 |
| No. of CTD casts | 40 | 8 |
| No. DIC/TA bottle samples | 260 | 67 |
| Mean (min. - max.) wind speed [m s$^{-1}$] | 10.7 (4-16) | 6.6 (3-10) |

[a] Two casts done on the 8$^{th}$ of January have not been included into this case study, as they were too disconnected from the rest of the time series. However, the $CO_2$ flux based on any discrete measurements available from these casts has been included in Fig. 8. ADCP measurements started on the 7th of January. The underway $f CO_2$ measurements started on the 7$^{th}$ of January at 23:00 (Fig. D1).

fugacity of $CO_2$ is numerically similar to the partial pressure of $CO_2$ ($pCO_2$), but accounts for the non-ideal behaviour of the gas. The $fCO_{2atm}$ is calculated using the virial- and cross-virial coefficients from Weiss (1974) in the equation to convert the atmospheric $CO_2$ mole fraction to $fCO_2$ by Weiss and Price (1980). It requires the dry air mole fraction of $CO_2$ ($xCO_2$), for which we use discrete air sample measurements from Syowa Station at 69° 0′ 16″S, 39° 34′ 54″E (Dlugokencky et al., 2019), and the water vapour pressure, which is derived from *in situ* seawater temperature and salinity in the parameterisation of Weiss and Price (1980). We use the average $xCO_2$ of January 2015 and 2019 for the PS89 and PS117 experiment, respectively, and shipboard atmospheric pressure reported at sea level.

The $fCO_2$ of surface seawater ($fCO_{2sw}$) is determined from the DIC, TA, nitrate, and phosphate content, as well as the temperature and salinity, of the shallowest discrete seawater samples (typically between 15 and 5 m deep), using PyCO2SYS (Version 1.3) (Humphreys et al., 2022), based on the original work of Lewis and Wallace (1998). Parameterisations by Lueker et al. (2000) were used for the carbonic acid equilibrium constants, by Dickson (1990) for the bisulfate ion dissociation constants, by Uppström (1974) for the boron:salinity relationship, and by Dickson and Riley (1979) for the hydrogen fluoride dissociation constants. Additionally, continuous surface water $fCO_2$ was measured at the ship's seawater supply with an intake at 11 m depth by a General Oceanics $pCO_2$ analyser on board, for the PS89 tidal observation only. The discrete surface seawater samples were collected at the same depth as the underway water intake depth ($\sim$11 m) or shallower ($\sim$5 m). The measured and calculated $fCO_2$ values are comparable for the periods where they overlap (Fig. D1). Small discrepancies between the measured and calculated values are likely due to the difference in depth and spatial and temporal variability. As is shown in Section 4.2, these discrepancies are not large enough to affect the agreement between the $CO_2$ flux estimates based on these two sets of $fCO_2$ data. Alongside $fCO_2$, the PyCO2SYS package simultaneously resolves other carbonate system

parameters with DIC, TA, and auxiliary data (listed above) as input parameters (Zeebe and Wolf-Gladrow, 2001; Humphreys and Matthews, 2022). These include the saturation state for calcite and aragonite (polymorphous forms of calcium carbonate) and pH.

## 3 Results

### 3.1 Tidal current

The $u$ (east-west) and $v$ (north-south) components of the current velocity measured by the ADCP are significantly positively correlated to each other (Fig. F1). They synchronously alternate sign roughly twice a day throughout the water column (Padman et al., 2002), resulting in a barotropic (depth-averaged) component of the flow that matches the modelled tidal velocity (Fig. 2). The semi-diurnal tidal currents are thus a dominant component of the total current velocity at this sampling location. The vertical grey areas in Fig. 2 (and in other figures) indicate the time and duration of ebbing tide, here defined as the time when both the $u$ and $v$ components of the modelled current velocity are positive, i.e. the direction of the current is towards the north-east. When both components are negative, the direction is towards the south-west and here considered to be rising tide. Note that the white vertical areas in the figures include the rising tide as well as times when either the modelled $u$ or $v$ component is negative.

The velocity profiles are generally homogeneous during ebbing tide. More vertical structure is seen in both velocity components after peak velocities are reached during rising tide during PS89. These baroclinic (depth-dependent) flows induce short-lived vertical shear between the surface and subsurface layers before velocity profiles homogenise again towards the north-east (ebb). The range of current velocities is slightly larger during the PS89 tidal observation than during PS117 (Fig. F1), which is mainly due to the higher velocities in the morning of 10 January 2015. The modelled tidal current at this time is not particularly stronger than at any other time point in the observation period, including PS117. The stronger current in the first half of 10 January 2015 is thus likely the result of other enhancing factors, such as wind speed and direction.

Both the PS89 and PS117 discrete tidal observations were made five days after spring tide, although the spring tide prior to the PS89 observations was at full moon (5 January 2015) and the spring tide prior to the PS117 observations was at new moon (6 January 2019; dates obtained from https://tidesandcurrents.noaa.gov/historic_tide_tables.html). The tidal observations of both case studies are thus set at a similar time in the spring-neap tidal cycle. Peak-to-peak tidal heights in this region are around 2 m, but may increase by a factor of two during spring tides (Padman et al., 2002, 2018). Our datasets only capture a small fraction of time, and we cannot judge the relative strength of the tide beyond its time limits. However, considering that the dominant constituent of tidal oscillations in the Weddell Sea is the semi-diurnal constituent (Padman et al., 2002) and that the modelled velocities are similar between both tidal observations, we regard the two case studies as comparable in terms of tidal influence on the system.

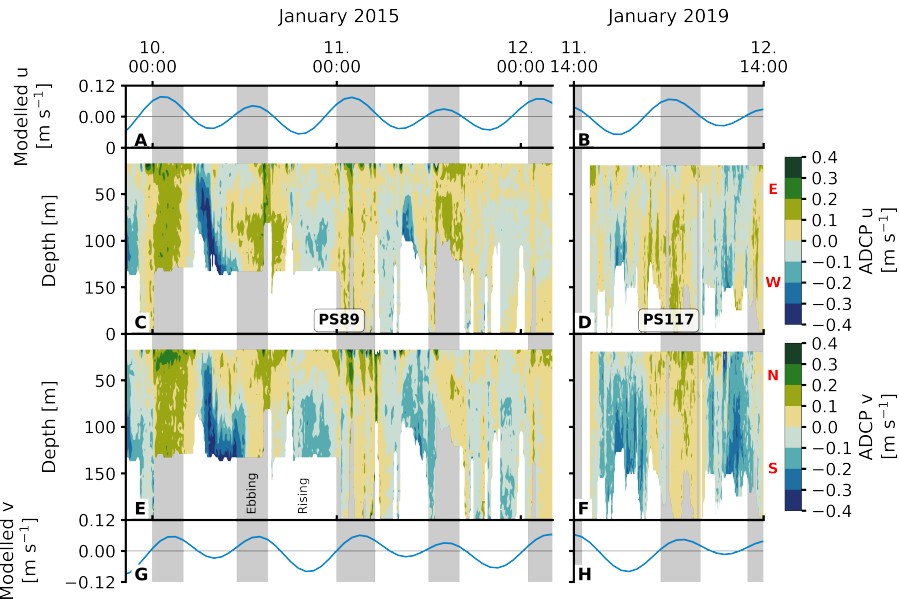

**Figure 2.** Current velocity for PS89 (A, C, E, G) and PS117 (B, D, F, H). Modelled *u* component of the tidal currents is shown in panels A and B. The modelled *v* component is shown in G and H. Modelled tidal velocities are averaged over the full water column. The *u* component of the ADCP profile data is shown in C and D. The *v* component of the ADCP data is shown in E and F. Vertical grey-shaded areas indicate times of ebbing tide, here identified as times when the *u* and *v* components in the modelled tidal current are both positive, i.e. the direction of the current is north-east. Directions associated to the positive and negative values of the *u* and *v* components are indicated by the red letters next to the colour bars: N (north), S (south), E (east), W (west).

## 3.2   Physical variability

A clear semi-diurnal tidal cycle is observed in the physical properties of the water column during the PS89 tidal observation period in January 2015 (Fig. 3). Over the depth horizons, salinity increases and temperature decreases throughout the water column during ebbing tide. Isopycnals rise to the surface, resulting in a shallow stratification. A mixed layer forms at the surface during ebbing tide with relatively uniform properties and an average depth of ∼20 m. The mixed layer depth (MLD) is determined by a 0.03 kg m$^{-3}$ density difference with the average of the top 10 m. During rising tide, the MLD tends to either deepen or the mixed layer breaks down altogether as the density decreases. When a steep density gradient is absent, which typically occurred during rising tide, the water column is considered to be relatively well mixed. The isolines for salinity and temperature deepen as the surface water becomes fresher and warmer. These semi-diurnal fluctuations occur throughout the water column, down to the bottom at almost 200 m depth.

While a similar tidal pattern is also recorded in the salinity and temperature profiles of the shorter PS117 tidal observation, the amplitude of variability for salinity is much smaller compared to PS89. The salinity values for PS117 range only between 34.13 and 34.29, while they range between 34.03 and 34.34 for PS89. With values between -1.58 °C (at the surface) and near

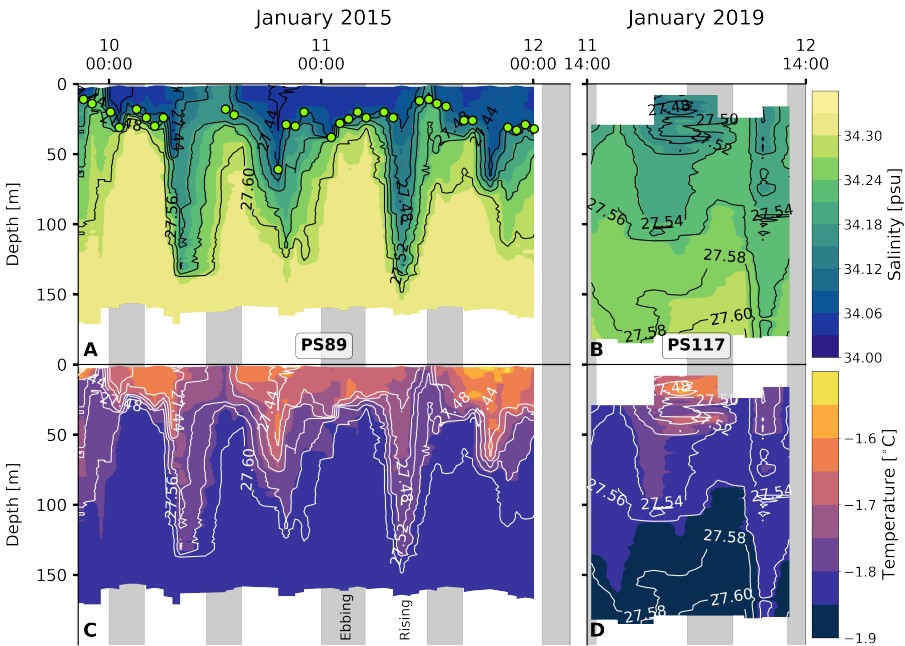

**Figure 3.** A and B show the salinity for PS89 and PS117, respectively. Green markers in A indicate the mixed layer depth (MLD) for casts where a mixed layer could clearly be identified according to a strong density gradient. MLD is identified as the depth at which the density difference with the average density values of the top 10 m is 0.03 kg m$^{-3}$. Temperature profiles are shown in C (PS89) and D (PS117). White contour lines represent isopycnals in kg m$^{-3}$. Vertical grey shaded areas indicate times of ebbing tide.

freezing -1.89 °C (at the bottom), the temperatures measured during PS117 have a smaller range and are generally lower than those measured for PS89 (-1.42 to -1.85 °C). In contrast to PS89, no mixed layer was identified at any point during the PS117 tidal period, as a steep density gradient was absent and the entire water column was better mixed.

### 3.3 Biogeochemical variability

Despite the lower vertical resolution compared to the continuous sensor measurements, the discrete bottle measurements capture the variability in the water column well, as shown by comparing the discrete bottle and continuous profile measurements for density (Fig. E1). We are therefore confident that the discrete biogeochemical measurements are also representative of the variability in the water column at the time of sampling.

The DIC and TA content mimic the tidal signature as seen for salinity (Fig. 4): their content decreases throughout the water column when the tide comes in, and increases as the tide goes out. The variability is, again, much lower during PS117 than during PS89. For the PS89 tidal observation, the DIC (TA) varied between 2174.3 and 2217.9 $\mu$mol kg$^{-1}$ (2307.3 and 2327.1 $\mu$mol kg$^{-1}$) at the surface (<50 m) and between 2186.8 and 2222.7 $\mu$mol kg$^{-1}$ (2312.3 and 2328.7 $\mu$mol kg$^{-1}$) at greater depths (>50 m). During PS117, DIC and TA content overall ranged from 2202.6 to 2220.1 $\mu$mol kg$^{-1}$ and 2311.5 to 2322.3 $\mu$mol kg$^{-1}$, respectively. In the ocean, changes to salinity are driven by oceanographic processes, such as dilution,

ice formation, and mixing. These physical processes also impact DIC and TA, which is why DIC and TA are often strongly correlated to salinity (Middelburg et al., 2020). This is also the case in our dataset. However, DIC and TA content are addition-
ally a function of biological (e.g. photosynthesis, respiration, and remineralisation) and chemical (e.g. $CaCO_3$ dissolution and precipitation) processes (Zeebe and Wolf-Gladrow, 2001). To be able to study the role of biogeochemical processes on DIC and TA content, it is useful to separate the effect of physical processes, such as dilution and mixing of different water masses, from the rest. This can be done by normalising the DIC and TA values to the salinity, which we have done here according to methods by Friis et al. (2003). The salinity-normalised DIC (nDIC) and TA (nTA) profiles lose much of the semi-diurnal vari-
ability seen in the profiles of the non-salinity-normalised values, suggesting that physical processes are the dominant drivers of the observed variability in DIC and TA (Fig. G1). PS117 nDIC values are markedly higher than those for most PS89 samples (Fig. 5). A part of this increase in nDIC over time could be result of the increase in atmospheric $CO_2$, assuming at least partial equilibrium with the atmosphere. The atmospheric $f\mathrm{CO}_2$ increase (10 $\mu$atm) alone could contribute $\sim$6 $\mu$mol kg$^{-1}$ to the surface DIC content if all other variables remained the same. This upper-bound estimate is based on average values of the top 10 m during PS89 and assumes equilibration of the surface water with the atmosphere.

PS117 nitrate (28.9 - 30.1 $\mu$mol kg$^{-1}$), phosphate (2.0 - 2.1 $\mu$mol kg$^{-1}$), and dissolved oxygen (322.0 - 333.2 $\mu$mol kg$^{-1}$) concentrations throughout the water column have similar values as those at the bottom of the water column during PS89 (Fig. 6, G2, G4). This is not the case for silicate, for which its PS117 values lie around the mean silicate values measured for PS89 (60.0 $\pm$ 1.1 $\mu$mol kg$^{-1}$; Fig. G2, G4). This observation illustrates that silicate behaves differently from phosphate and nitrate, as its content is affected by other processes, i.e. by diatom growth and remineralisation at depth or in the sediment rather than by photosynthesis and biological respiration (Sarmiento, 2013). Similarly to nDIC and nTA, the nutrients were salinity-normalised (following Friis et al. (2003)). Consistent to nDIC and nTA, no obvious deviations from the mean salinity-normalised values are observed that could indicate a dominant biological influence (Fig. G5). It supports the observation made above that the dominant driver of the variability observed within each tidal case study is mostly physical. For the salinity-normalised silicate content, the averages of both tidal observations are similar to each other (59.5 $\pm$ 0.5 $\mu$mol kg$^{-1}$ for PS89 and 59.2 $\pm$ 0.2 $\mu$mol kg$^{-1}$ for PS117; Fig. G5), indicating that processes affecting silicate content did not differ much between the two case studies. However, a consistent offset between the case studies is observed for the salinity-normalised values for nitrate and phosphate, where PS117 values are on average higher by 1.6 $\mu$mol kg$^{-1}$ and 0.1 $\mu$mol kg$^{-1}$, respectively. Even though biogeochemical processes might not be able to explain the variability within each tidal observation, these results suggest that the polynya may have had a higher input and/or a lower loss of nitrate and phosphate during January 2019 compared to January 2015. Fluorescence is here used as a proxy for the presence of photosynthetic cells. While the rising tide increased fluorescence in the water column during PS89, it was barely detected during PS117 (Fig. 6). In the absence of active photosynthetic cells, remineralisation enhancing the nitrate and phosphate content may have been an important process in the water observed during PS117.

Due to complications with the data for dissolved oxygen and fluorescence on four of the CTD casts of the PS117 tidal observation, these data were excluded from analysis. The implication of this reduced temporal resolution is that we risk losing representation of water column variability in our dataset for these two variables. However, the low water column variability

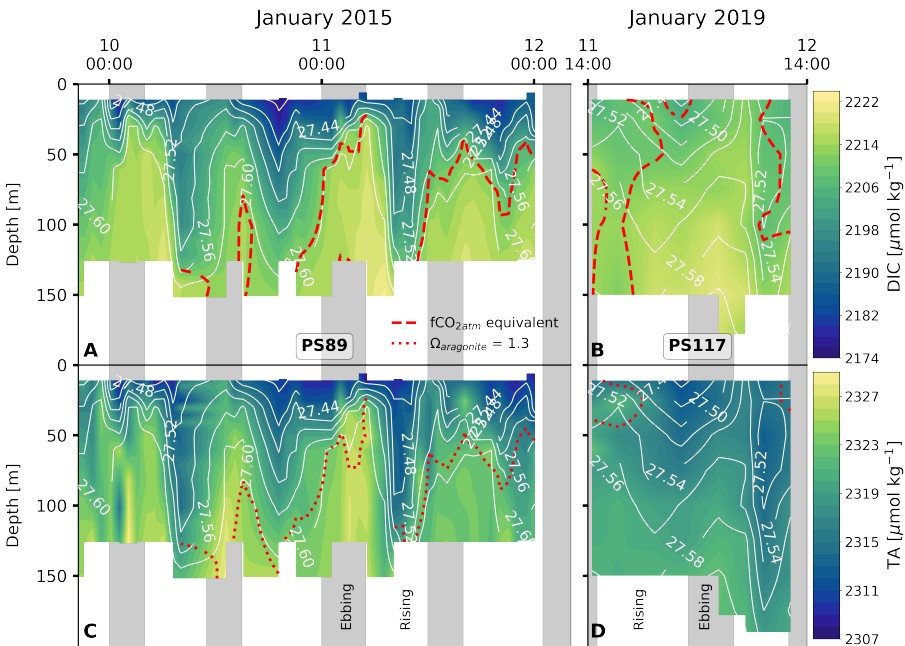

**Figure 4.** DIC content at the sampling site during the PS89 (A) and PS117 (B) tidal observations. TA content for the PS89 (C) and PS117 (D) observations. White contour lines indicate sigma-t in kg m$^{-3}$. Vertical grey shaded areas indicate periods of ebbing tide, as defined in the text. Red dashed line in A and B represents the depth at which the seawater $f$CO$_2$ is equal to the atmospheric $f$CO$_2$, which is 377 $\mu$atm in January 2015 and 387 $\mu$atm in January 2019. Seawater shallower than this depth is undersaturated in $f$CO$_2$ compared to the atmosphere. The red dotted line in C and D represents the depth at which $\Omega_{ar} = 1.3$. $\Omega_{ar}$ is lower at depths below this line. The value 1.3 is arbitrary to illustrate the vertical variability in the water column. $\Omega_{ar}$ ranges between 1.22-1.52 and 1.21-1.34 for PS89 and PS117, respectively.

over time for the physical and other biogeochemical variables (for which we do have data from every cast) strongly suggest that there likely is not a lot of variability in the dissolved oxygen or fluorescence content that we are missing out on due to
245 missing profiles (Fig. 6).

## 4 Discussion

### 4.1 Water masses and biogeochemistry

Hydrographically, the water measured during both tidal observations is identified as Eastern Shelf Water (ESW), which is found south of the Antarctic Slope Front (ASF) where pycnoclines slope downwards towards the south at the edge of the
250 narrow continental shelf of the Dronning Maud Land coastline (Heywood et al., 1998). ESW is characterised by salinities between 34.28 and 34.4 and temperatures close to freezing point (Carmack, 1974). ESW itself is a mixture of the following (Fahrbach et al., 1994):

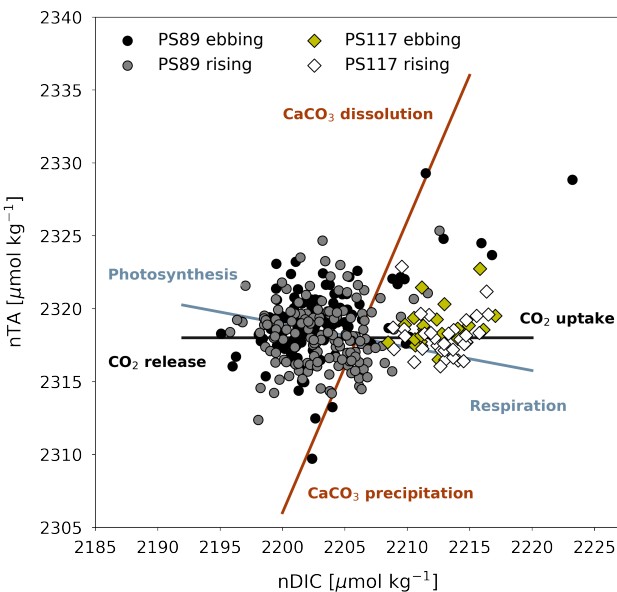

**Figure 5.** Salinity-normalised TA (nTA) and DIC (nDIC) plotted against each other for PS89 (circles) and PS117 (diamonds) tidal data. Samples collected during ebbing (black circles and yellow diamonds) or rising tide (grey circles and white diamonds) are differentiated by different marker colours. Theoretical process lines are drawn for $CaCO_3$ dissolution/precipitation, photosynthesis/respiration, and $CO_2$ uptake/release (Zeebe and Wolf-Gladrow, 2001).

- Winter Water (WW, winter surface water capped by a warmer summer stratification; characterised by a subsurface temperature minimum (Nicholls et al., 2009))

- Antarctic Surface Water (AASW, derived from WW that has been freshened by sea ice melt and heated by solar radiation in the spring and summer)

- modified Warm Deep Water (mWDW, which is the result of mixing between WW and WDW along the ASF (Ryan et al., 2020))

- glacial meltwater (GMW, formed by melting of glaciers, and of ice shelves induced by intrusions of warmer water, such as mWDW and AASW, underneath the ice shelf (Fahrbach et al., 1994; Zhou et al., 2014)).

However, the oscillations in the physical and biogeochemical properties induced by the incoming and outgoing tide suggests that the tide is enabling movement and possibly mixing of water masses, which is especially pronounced in the PS89 dataset (Fig. 3). While water in the Ekström polynya can broadly be categorised as one water mass (i.e. ESW) according to the physical properties, we will here explore the deepening and shoaling of the isopycnals, as well as the changing water physico-chemical properties.

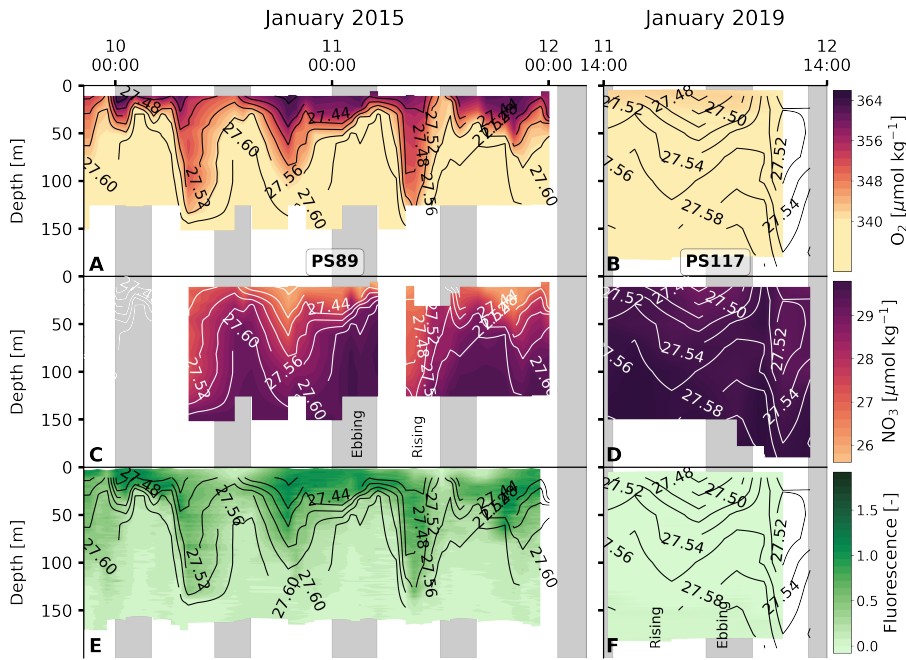

**Figure 6.** Same as Fig. 3, but for dissolved O$_2$ (A and B), NO$_3^-$ (C and D), and fluorescence (E and F). Fluorescence during PS117 varied at values < 0.09.

At rising tides during PS89, the tide brings in water from the north-east that is fresher and warmer compared to the water present at the sampling location during ebbing tide (Fig. 3 A, C). A well-defined mixed layer with relatively uniform properties is seen at the surface during ebbing tide ($\sim$ 20 m depth), which tends to disappear or deepen when the tide comes in and the water column density decreases. Similar observations of tide-driven shoaling and deepening of isopycnals have been made in

other Antarctic coastal systems, although in lower wind conditions (Llanillo et al., 2019, <4 m s$^{-1}$, see Table 1 for PS89 wind speeds). A destabilising (stabilising) water column has previously been associated with the rising (ebbing) tide observed in coastal polynyas in the Arctic (Skogseth et al., 2013). In the latter study, the tidal variability was characterised by a salinity front that moved back and forth with the tide. This "salinity front" is characterised by lower and higher salinities in the water column on either side of a sharp horizontal salinity gradient. A salinity front of this description might have been moving

back and forth with the tide over the sampling site in the Ekström polynya, which from a Eulerian perspective resulted in the properties of the water column changing as shown in Fig. 3 and 4.

In our case study, the fresher water with lower DIC and TA content on the north-eastern side of the salinity front (and sampling site) is likely advection of AASW, influenced by summer sea ice melt. Based on the fluorescence increase during rising tide for PS89 (Fig. 6E), the water on this side of the front seems to be richer in phytoplankton cells compared to the

280 south-western side of the front. In addition to a dilution effect, the accompanying increased fluorescence signal during rising tide suggests that photosynthesis in this water has likely contributed to its lower DIC, TA, and nutrient content (Fig. 4, 6),

which is sustained by solar radiation. The advection of fresher, lower DIC and TA seawater from the north-east during rising tide is likely enhanced by high winds (4 - 16 m s$^{-1}$) that consistently came from the north-east during - and in the week prior to - the PS89 tidal observation (Fig. H1A and H2A ).

In comparison to PS89, the sampling site during PS117 has less pronounced freshening of the surface water during rising tide, thereby highlighting the interannual variability of the system. Instead, the entire water column is much more uniform with very few vertical gradients during the observation period. Differences in shape and size of the polynya between the two case studies may have affected the proximity of a salinity front to the sampling site. Even though the average sea ice concentration in the immediate vicinity of the sampling site was highly variable in the two months leading up to both PS89 and PS117 (not

shown), the polynya was at least three times larger and more well-defined during PS117 than during PS89 (Section 2.1, Fig. A1 and A3). If a salinity front existed during PS117, it could have been located further away from the ice shelf edge, and the sharp horizontal salinity gradient might therefore not have passed directly over the sampling site during PS117 as it did during PS89. Moreover, winds during PS117 may not have had the same enhancing effect on the advection of fresher, diluted water during rising tide compared to PS89. Whereas high winds during PS89 consistently came from the north-east, the wind speeds during

PS117 were lower (3 - 10 m s$^{-1}$) and their direction was more variable, mainly coming from the ice shelf in the south-east (Fig. H1B and H2B). Modulating effects of winds on tidally-induced changes in physical water column properties, whether enhancing or counter-balancing, have also been observed in coastal systems at the Antarctic Peninsula (Llanillo et al., 2019).

    The other side of the salinity front (i.e from the south-west) brings in higher salinity and lower temperatures into the polynya during ebbing tide. This water is less ventilated than AASW and shares physical properties with ice shelf water. Ice shelf water

is characterised by potential temperatures of <-1.8 °C and salinity >34.6 (Carmack, 1974). However, recent work by Smith et al. (2020a) has shown that the temperature and salinity from a number of CTD profile measurements underneath the Ekström Ice Shelf in the same summer season as PS117 (2018/2019) ranged from -2.08 to -1.83 °C and 34.21 to 34.38, respectively (Fig. 7). These values overlap with the temperature and salinity measured within ~10 m of the bottom of the sampling site during both tidal observations, especially when the tide goes out.

It is possible that the ebbing tide draws out water from underneath the ice shelf which is expected to be colder. Indeed, this possibility is supported by findings in Smith et al. (2020a), which includes a repeat profile of the Ekström Ice Shelf's cavity water at one of the measurement stations on the ice shelf (EIS-4; Fig. 7). The two repeat profiles at EIS-4 were taken 11 hours apart. The small difference observed in the vertical salinity and temperature profile between these two casts was attributed to tidal influences extending underneath the ice shelf (Smith et al., 2020a). As well as some vertical displacement of the ice shelf

by the tidal force (Legrésy et al., 2004), horizontal displacement also occurs underneath the ice shelf. Another study found 12-hourly and 14-day fluctuations in a temperature time series 70 m underneath the Ekström Ice Shelf (June 2012 to February 2013), ranging between -1.94 and -1.6 °C (Hoppmann et al., 2015). It also described effects of the tide on the glacial meltwater outflow, which could be seen in the orientation and growth of platelet ice crystals in Atka Bay (Hoppmann et al., 2015), adjacent to the Ekström Ice Shelf and east of the sampling site in the current study. In terms of biogeochemical properties, the sub-ice

shelf water is expected to be less ventilated compared to the AASW and to have relatively high nutrient and DIC content, and lower dissolved oxygen content, as a result of net remineralisation and lack of exchange with the atmosphere. Due to the

lack of exposure to the atmosphere, phytoplankton cells (for which we use fluorescence as a proxy) are expected to be mostly absent. This description of water properties is consistent for the properties observed during ebbing tide (Fig. 4, 6). It therefore seems feasible that less ventilated, colder water from underneath the ice shelf with lower oxygen and higher nutrient and DIC content can extend to the edge of the ice shelf during ebbing tide and into the polynya.

Even though mWDW is a source for eastern shelf water (Nicholls et al., 2009), we do not see a direct signal of mWDW in our dataset. The bathymetry of the cavity underneath the Ekström Ice Shelf slopes from about 450 m depth at the ice shelf edge to a maximum depth of 1100 m southwards towards the grounding-line (Smith et al., 2020a). Although troughs sculpted into the cavity's bathymetry (Smith et al., 2020a) potentially allow WDW (or its modified form) to enter it (Fahrbach et al., 1994) (a process that may be enhanced by tides), the warmer water is usually restricted to the north of the continental shelf in this region by prevailing easterly winds (Heywood et al., 1998; Thompson et al., 2018). The cavity is instead likely influenced by ESW that enters through Ekman transport (Zhou et al., 2014).

As explained in Section 3.3, the salinity normalisation removes the impact of physical processes from the DIC and TA data. Therefore, any variability that remains in the nDIC and nTA results needs to be explained according to other processes. These processes are represented in Fig. 5 by theoretical lines that indicate how nTA and nDIC would change relative to each other as a result of photosynthesis/respiration, $CaCO_3$ dissolution/precipitation, and $CO_2$ uptake/release (Zeebe and Wolf-Gladrow, 2001). For example, factors that could be relevant to net photosynthesis are variable sea ice cover affecting light availability, nutrient replenishment during ebbing tide, and mixing of phytoplankton cells into deeper water during rising tide (Gleitz et al., 1994). Yet, none of these processes seem particularly dominant in changing the nTA and nDIC content within each tidal observation (Fig. 5). The results in Fig. 5 show a legacy of processes that may have occurred in the weeks to months prior to sampling, as the marine carbonate system's equilibration time with the atmosphere is slow, especially in sea ice covered regions. Additionally, the data during rising tide might also reflect processes that happened in the sea ice, which will have affected the carbonate chemistry of the sea ice meltwater and thus the properties of the AASW. While we here consider the tides to transport a salinity front back and forth across the sampling site, we must also recognise that the sampled mass of water on each side of the front is not exactly the same during each tidal phase. This contributes to the variability observed in the dataset. Finally, even though there is good agreement between the high vertical resolution sensor data and the discrete bottle data for salinity and temperature (exemplified with density in Fig. E1), biogeochemical processes could imprint additional variability in the DIC and TA profiles that are not reflected in salinity and temperature measurements. We must therefore consider that the discrete seawater samples might not have captured the full scale of the variability in the polynya, limiting our interpretation of relevant biogeochemical processes.

Coastal polynyas have been described as "the most productive waters in the Southern Ocean" (Arrigo et al., 2015). Their chlorophyll *a* levels are found to peak in January, roughly coinciding with the peak of coastal polynyas' area of open water (Arrigo et al., 2015). In polynyas around Antarctica, iron supplied by basal melting of ice shelves is a major contributing factor to the variability in phytoplankton biomass (Arrigo et al., 2015). Considering that the observations in the current study were made in a coastal polynya of variable size directly adjacent to the Ekström Ice Shelf with an estimated basal melt rate of 4.2 Gt year$^{-1}$ (Rignot et al., 2013), the lack of observable *in situ* nutrient and inorganic carbon uptake by primary

productivity at the surface is perhaps unexpected. This also applies to times at rising tide in the PS89 observation period when the fluorescence signal increases in the water column, suggesting advection of phytoplankton cells into the polynya. Tidal mixing that replenishes nutrients at the surface has been described to drive phytoplankton productivity in other tidal regions (Rogachev et al., 2001). However, primary productivity and its peak in the summer have been shown to be highly variable among Antarctic coastal polynyas, and important drivers of low primary productivity are deep MLD and grazing pressure (Arrigo and van Dijken, 2003). These are likely highly relevant factors in our case study. High phytoplankton growth rates require water column stability that lasts several weeks (Gleitz et al., 1994), which is absent during the case studies presented here. Even during ebbing tide, when the water column stratifies at a shallow depth during PS89, the duration of stratification is too short to support substantial *in situ* DIC and nutrient uptake by growth and primary production.

In a study in the Amundsen Sea, stations close to the Pine Island Glacier were characterised by a deep MLD and low phytoplankton biomass and Chl *a* despite high dissolved Fe availability (Alderkamp et al., 2012), suggesting that upwelling of Fe-rich basal meltwater mixed the water at the ice shelf's edge (Alderkamp et al., 2012; Gerringa et al., 2012). A colder, more buoyant layer along the base of the Ekström Ice Shelf similarly indicates outflows of ice shelf meltwater (Smith et al., 2020a). The temperature and salinity at the bottom of the profiles during PS117 compare to those measured near the base of the ice shelf (mean of top 350 m of the ice shelf CTD casts EIS4-8 (Smith et al., 2020a)). The bottom water temperature and salinity values for PS89 resemble more closely those deeper in the cavity (e.g. the mean of the water deeper than 350 m of the ice shelf CTD casts EIS4-8; Fig. 7). Earlier, we noted that southerly winds during PS117 may have counter-acted some of the advection of fresher, more ventilated water from the north-east during rising tide. The comparisons to the work by Smith et al. (2020a) support the idea that - in addition to less sea ice meltwater input from the north-east - outflow and mixing of ice shelf meltwater might have been stronger during PS117 than PS89, dominating the polynya water properties. Along with a more mixed water column, this difference in connectivity to the ice shelf cavity is consistent with the less ventilated water of the polynya during PS117, and can be responsible for the higher nDIC (and salinity-normalised nitrate and phosphate) content, compared to PS89.

From an ecological perspective, it is relevant to consider the effect of the carbonate system variability on the diversity, structure, and production of pelagic-benthic organism communities. Calcifying organisms, such as pteropods, foraminifera, and coccolithophores, depend on the seawater calcium carbonate saturation state to form their shells and skeletons, which are made from $CaCO_3$ (Orr et al., 2005). The pH in the Ekström polynya varied between 8.02-8.12 and 8.02-8.06 for PS89 and PS117, respectively (see Fig. G1 for vertical variability). The saturation state of aragonite ($\Omega_{ar}$; the less stable polymorph compared to the other common $CaCO_3$ polymorph: calcite) concurrently varied between 1.22-1.52 and 1.21-1.34, respectively. A contour in Fig. 4C and D at an arbitrary value of 1.3 for $\Omega_{ar}$ gives a sense of the vertical variability driven by tides. Even at the lowest pH values recorded here, the $\Omega_{ar}$ does not fall under 1, which means that the marine chemical environment does not thermodynamically promote $CaCO_3$ dissolution. The dynamic nature of the polynya might foster a resilience among the pelagic and benthic organism communities to rapid (semi-diurnal) changes of $\Omega_{ar}$. However, even at a carbonate saturation level > 1, the rate of biogenic calcification has been shown to be affected by the $CaCO_3$ saturation state (Feely et al., 2004). The Southern Ocean is especially vulnerable to ocean acidification driven by marine anthropogenic $CO_2$ uptake (Orr et al., 2005; Negrete-García et al., 2019). Tidally induced variability may increase the sensitivity of high-latitude coastal systems to shoaling

aragonite and calcite saturation state horizons. Alongside ecological impacts of tides along the Antarctic coastline, future studies can look into the tidal impacts on long-term changes of the vulnerability of pelagic-benthic organism communities.

In this study, we have argued that the DIC variability in the coastal polynya is driven by back-and-forth movement of water under the force of tidal currents across the sampling site located in a region where there is a horizontal gradient in DIC content: lower DIC content to the north-east, influenced by summer sea ice melt, and higher DIC content to the south-west, influenced by unventilated ice shelf cavity water. This led us to investigate whether there is evidence in our dataset for a tidally-driven horizontal DIC pump. For example, net transport away from the ice shelf could transport DIC and nutrients (and perhaps even iron) from the ice shelf towards surface waters on the continental shelf that are exposed to sea ice and the atmosphere. Subsequent biological carbon uptake will then remove DIC. However, when we calculate the trajectory of a water parcel (using the ship's position as a starting point and the average current velocity of the water column) the net transport is south/south-east, i.e. towards the ice shelf (Fig. F2). This implies that surface waters would be modifying the properties of the water underneath the ice shelf over time (instead of the other way around), for example by dilution. If this is the case, we would expect to see a trend in the DIC content of the polynya during ebbing tide. However, this is not the case and the net change in DIC content over six hour periods (including ebbing and rising tide) is zero (Fig. G3 for PS89). Our observations are a snapshot of a highly dynamic system and consequently they do not provide enough data to analyse such modifications and trends of the seawater physico-chemical properties. Nevertheless, they can be the beginning of future studies into this topic.

While a tidal movement of a salinity front or gradient across the sampling site is a possible and realistic explanation for the oscillations observed in the water column, validating it would require physico-chemical profiles between the Ekström Ice Shelf and the edge of the continental shelf. Without them, other relevant physical processes, such as regional circulation, tidal straining, or mixing, cannot be excluded. For the same reason, the distance from the ice shelf at which the tidally-driven deepening and shoaling of the isolines (most clearly seen for PS89) can be observed remains uncertain. The distance will depend on the strength of the tidal current and would have to be investigated with repeat transects from the ice shelf towards the open ocean at various points in the tidal cycle. Other mediating factors, such as winds and internal tidal waves generated by uneven bathymetry, would have to be additionally considered (Llanillo et al., 2019). Moreover, tidal mixing may erode fronts and modify water masses (such as at the ice shelf edge or of the ESW) over time, adding a temporal dimension to the effect of the tide on the shelf waters in this region.

Despite the unknowns outlined above, the case studies presented in this work show that strong tidal influences on the physical structure and biogeochemical properties of the water column can be expected along the Weddell Sea coastline (and other polar regions subject to strong tides), especially in close proximity to ice shelves and regions of sea ice melt. They also show that local winds and ice shelf meltwater outflows can increase the complexity of the tidal impact within in a region such as a coastal polynya. In addition to studies on the physical role of tides on (for example) basal ice shelf melt, ecological, biogeochemical, and air-sea gas exchange studies can benefit from a better understanding of tidal impacts on the water column.

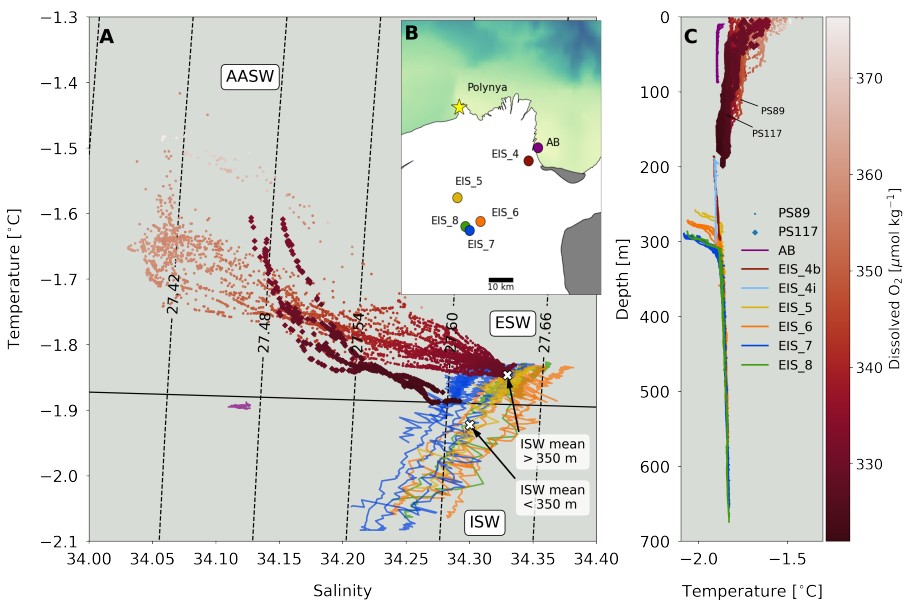

**Figure 7.** A) Temperature-salinity diagram for PS89 (circles) and PS117 (diamonds) tidal observation periods, which are coloured according to dissolved oxygen concentrations. CTD profiles of the ice shelf's cavity water were collected and made available by Smith et al. (2020a) (coloured lines). The cavity CTD profiles were taken by hot water drilling through the ice at various locations on the ice shelf, which are shown on the map in (B) in corresponding colours to the profiles in (A). AASW = Antarctic Surface Water, ESW = Eastern Shelf Water, ISW = Ice Shelf Water. The average temperature and salinity of water deeper and shallower than 350 m underneath the ice shelf are plotted as single white crosses. Contour lines indicate sigma-t in kg m$^{-3}$. Black line indicates freezing point at mean atmospheric pressure. B) Map of measurement locations of the Ekström Ice Shelf cavity CTD profiles by Smith et al. (2020a) denoted by "EIS_", a measurement location in Atka Bay by Smith et al. (2020a) denoted with "AB", and the sampling location of the tidal observations indicated by the yellow star. C) Temperature profiles for PS89 (circles) and PS117 (diamonds) tidal observation periods, coloured according to oxygen concentrations, and for the hot water drill CTD profiles underneath the ice shelf from Smith et al. (2020a).

## 4.2  Air-sea CO$_2$ exchange

The tidally-induced variability in the water column implies that the timing of sampling matters when estimating the relative contribution of coastal regions to the total CO$_2$ flux of the Weddell Sea. We first present the variability in the CO$_2$ flux during the tidal observation periods. We then illustrate that bias can be introduced in our understanding of the relative importance of coastal regions along the Weddell Sea if sampling strategies do not take tidal influences into account.

The difference in the $f$CO$_2$ of the seawater and the atmosphere is what drives a positive or negative CO$_2$ flux at the air-sea
interface. For the flux calculations, we used the average wind speed recorded during the PS89 and PS117 tidal observation periods: 10.7 and 6.6 m s$^{-1}$, respectively (Table 1). This choice implies that the computed variability in the flux results is mainly driven by the air-sea gradient of $f$CO$_2$. During the PS89 tidal observation period in January 2015, the surface water at the Ekström Ice Shelf was undersaturated in $f$CO$_2$ relative to the atmosphere (Fig. 4), creating a CO$_2$ sink (Fig. 8). However, the

depth at which the $f\mathrm{CO_2}$ of the seawater is equal to that of the atmosphere (marked by a dashed line in Fig. 4A) fluctuates from
near the bottom of the water column during incoming tide to near-surface during outgoing tide. Direct $f\mathrm{CO_2}$ measurements
were made using the vessel's underway system (at 11 m depth) during (as well as 2.5 days prior to the start of) the discrete
seawater sampling, i.e. the start of the PS89 tidal observation period (Fig. D1). $f\mathrm{CO_2}$ correlates very well with the salinity,
again indicating that physical movement of water is the dominant driver of the variability. $\mathrm{CO_2}$ fluxes determined from the
underway measurements a) compare well with the fluxes calculated from the discrete seawater carbonate chemistry results
where they overlap in time, and b) show an even stronger fluctuation on the $7^{th}$ and $8^{th}$ of January 2015, when the flux status
of the sampling site swung between a sink (rising tide) and a source (ebbing tide) of $\mathrm{CO_2}$ twice within 24 hours (Fig. 8). This
suggests that water with higher DIC content reached the surface during ebbing tide. The $\mathrm{CO_2}$ sink during PS89 is as large as
-23.6 mmol m$^{-2}$ day$^{-1}$, while the largest $\mathrm{CO_2}$ source reaches 3.1 mmol m$^{-2}$ day$^{-1}$ (Fig. 8).

During PS117, the seawater $f\mathrm{CO_2}$ also dips below the atmospheric value at times of rising tide. However, the $f\mathrm{CO_2}$ gradient
and the wind speed are much lower compared to PS89 (Fig. 4B), resulting in a low $\mathrm{CO_2}$ flux (Fig. 8). Even though not all PS117
CTD cast measurements started as shallow as those from PS89, the few casts that did have measurements starting $<20$ m
showed a lack of a strong gradient in the DIC and TA content at the surface. The shallowest discrete carbonate chemistry water
samples are therefore considered to be representative enough of the water properties at the surface. Accordingly, the site is
likely to have had a relatively neutral $\mathrm{CO_2}$ flux at this time in January 2019. Using the $f\mathrm{CO_2}$ results based on the discrete
water sampling, the average $\mathrm{CO_2}$ uptake during the PS89 tidal observation period is -11.7 $\pm$ 3.7 mmol m$^{-2}$ day$^{-1}$ ($\pm$ 1$\sigma$). For
PS117, the average $\mathrm{CO_2}$ release to the atmosphere is -0.1 $\pm$ 0.9 mmol m$^{-2}$ day$^{-1}$. Even though the size of the coastal polynya
was variable, the sampling site was free of sea ice during the tidal observations and it is assumed to have been ice free for the
entire summer. We therefore have not scaled the gas transfer velocity with the fraction of open water area. At times during the
summer when the site does get covered by sea ice, the amplitude of the $\mathrm{CO_2}$ flux would be subdued.

The importance of strategic seawater sampling for the purpose of obtaining reliable air-sea gas exchange values in regions
such as the Ekström coastal polynya is illustrated by the following. Assuming the scenario we know for the PS89 tidal obser-
vation period, if discrete water samples had unknowingly only been collected during ebbing tide (higher seawater $f\mathrm{CO_2}$), the
calculated $\mathrm{CO_2}$ flux (-3.1 mmol m$^{-2}$ day$^{-1}$) would have underestimated the strength of the $\mathrm{CO_2}$ uptake by the polynya by up
to 73 %, compared to the average uptake (-11.7 mmol m$^{-2}$ day$^{-1}$). However, if samples had only been collected during times
of rising tide (lower seawater $f\mathrm{CO_2}$), the capacity of the polynya to take up $\mathrm{CO_2}$ (-19.6 mmol m$^{-2}$ day$^{-1}$) would have been
overestimated by up to 67 %, compared to the average uptake value. Since the variability of the $\mathrm{CO_2}$ flux during PS117 was
much lower (a $\mathrm{CO_2}$ release ranging between -1.2 and 0.8 mmol m$^{-2}$ day$^{-1}$), samples collected at any time during this 24 hour
period would have been relatively representative of this tidal observation period, but not necessarily of the month or the entire
summer season.

We emphasise the potential misrepresentation of the role of coastal polynyas in the Weddell Sea $\mathrm{CO_2}$ uptake if tidal influ-
ences are not accounted for. For this, we again use the two extreme scenarios based on the PS89 observations that were also
used above to illustrate the maximum potential over- and underestimation of the $\mathrm{CO_2}$ uptake. I.e., we use the hypothetical cases
where surface seawater samples are either collected at peak rising tide (overestimation of $\mathrm{CO_2}$ uptake) or at peak ebbing tide

(underestimation of $CO_2$ uptake). We constrain results and comparisons to the summertime, assume that all coastal polynyas in the south-eastern Weddell Sea are influenced by the same water masses present in the Ekström polynya during the PS89 tidal observation, and assume the same wind speed to highlight the role of the seawater $f$CO$_2$ variability. Using the total area of polynyas along the south-eastern Weddell Sea coastline in the summer of $49 \times 10^3$ km$^2$ (as estimated by Arrigo and van Dijken (2003)), the total net $CO_2$ uptake for all polynyas along the south-eastern coastline would be -0.97 $\times10^9$ mol day$^{-1}$, if data had only been collected during rising tide. It would have been -0.15 $\times10^9$ mol day$^{-1}$, if data had only been collected during ebbing tide.

Brown et al. (2015) estimated a summer marine uptake of $CO_2$ for the entire Weddell Sea of -0.044 to -0.058 $\pm$ 0.010 Pg C yr$^{-1}$, based on a summertime ocean inversion. In our hypothetical, biased upscaling case of collecting carbonate chemistry samples during ebbing tide, we would determine that the contribution of eastern shelf polynyas to Brown et al.'s upper summer $CO_2$ uptake estimate is 1.2 % (using the same area for the Weddell Sea: $6.2 \times 10^{12}$ m$^2$ and scaling the daily flux up to the whole year). If seawater samples had only been collected during rising tide, the contribution of the eastern shelf polynyas would have been estimated at 7.3 % to the total summer $CO_2$ uptake of the Weddell Sea. This is quite substantial considering that the area used here for the south-eastern coastal polynyas is less than 0.8 % of the total Weddell Sea area (including regions covered and not covered by sea ice). If the average $CO_2$ flux of PS89 had been used in this simplistic upscaling exercise instead of the extreme high- and low-end scenarios, then the total $CO_2$ flux of all Weddell Sea coastal polynyas would be -0.58 $\pm$ 0.18 $\times10^9$ mol day$^{-1}$ (4.4 % of Weddell Sea flux) for January 2015. Results are two orders of magnitude lower when the same upscaling exercise is done with the average flux for the PS117 case study: 0.003 $\pm$ 0.034 $\times10^9$ mol day$^{-1}$ (0.02 % of Weddell Sea flux).

The purpose of the above exercise is to simply highlight the variability in these coastal systems. Given both the scarcity of data in these regions and the challenges in reaching them, tides may be an important aspect to consider to explain some of the variability seen in previous and future oceanographic data. Although the Weddell Sea is considered to be an - albeit small - annual net $CO_2$ sink (Hoppema et al., 1999; Bakker et al., 2008; Brown et al., 2015), its $CO_2$ uptake is sensitive to the balance of physical, chemical, and biological processes, such as sea ice growth/melt, regional wind strength/patterns, circulation, and biological $CO_2$ drawdown (Brown et al., 2015). The coastal marine regions and the processes that govern their water properties may be equally sensitive and their changes on time-scales of hours to weeks or months may be mediated by the tides.

# 5 Conclusions

We present the significant semi-diurnal influence of tides on the water properties and carbonate chemistry at the margins of a coastal polynya hugging the Ekström Ice Shelf in the south-eastern Weddell Sea. Advection of lower DIC and TA waters from the north-east during rising tide influenced by sea ice melt, decreases the salinity and $f$CO$_2$ at the sampling site and results in $CO_2$ draw-down from the atmosphere. As the sampling site is located directly next to the ice shelf, it sees the extension of the water underneath the ice shelf, which is drawn out from underneath during ebbing tide. This water is less ventilated and has

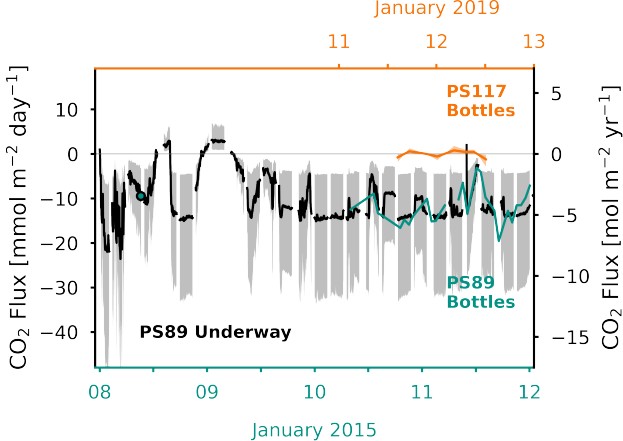

**Figure 8.** Air-sea $CO_2$ flux (in mmol m$^{-2}$day$^{-1}$ on the left y-axis and in mol m$^{-2}$year$^{-1}$ on the right y-axis) determined from the discrete surface seawater sample measurements and average wind speed for the PS89 (green, bottom x-axis) and PS117 (orange, top x-axis) tidal observation periods, and from the PS89 underway $fCO_2$ measurements (black, bottom x-axis), which started on the $7^{th}$ of January 2015. The $CO_2$ flux based on a discrete surface seawater sample collected on the $7^{th}$ of January 2015 is also shown with a green marker. The filled shading indicates the range of the flux calculated using the minimum and maximum wind speed measured during PS89 and PS117, respectively (Table 1). Negative flux represents $CO_2$ uptake by the ocean.

a higher DIC and TA content (and higher $fCO_2$) compared to the water to the north-east, which decreases the strength of the $CO_2$ sink and can even reverse the direction of the $CO_2$ air-sea flux on a semi-diurnal time scale.

Differences in the variability between the two tidal observations between January 2015 and January 2019 suggest a complex interaction between timing of the tide, local and regional sea ice melt, polynya area, basal melt, and local forces, such as wind
speed and direction. The datasets of the two short case studies presented here are too small to fully explore the modulating effects of these processes on the water column variability. To be able to do so, longer tidal observations are required that cover different parts of the spring-neap tidal cycle, and at different times of the year to capture varying wind and ice melt/growth conditions. Alongside carbonate system state variables, an array of co-collected measurements, such as micro-nutrients, biological productivity, and oxygen stable isotopes, can help to constrain interacting processes. An understanding of the carbonate
chemistry of the cavity water underneath the ice shelf - although challenging to obtain - would help understand the influence of this water on the polynya during ebbing tide.

The observations presented here were obtained from a stationary sampling site. Without knowing the hydrological conditions in the surrounding area at various time points in the tidal cycle, we must consider that our conclusions only apply to a very local area, the margins of polynyas, or the edges of ice shelves. Hydrographic transects between the ice shelf and well into the sea ice
cover at ebbing and rising tide can help identify the extent to which the tidal oscillation, as seen in the case studies, is relevant further away from the coastline and from the polynya's margins. It may also help identify the formation and characteristics of a horizontal coastal salinity gradient – here referred to as a "salinity front" – that moves back and forth with the tide.

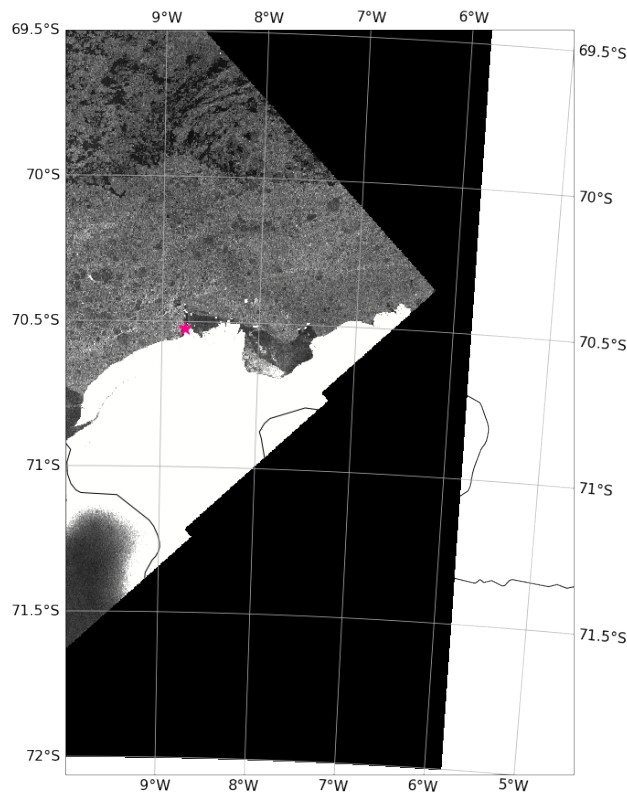

**Figure A1.** Sentinel-1 SAR image of the Ekström Ice Shelf and polynya on 12.01.2015 (PS89) (ESA, 2021). The pink star indicates the sampling location for the PS89 tidal observation.

This case study has shown that tides can swing the status of coastal polynyas on the south-eastern continental shelf of the Weddell Sea from a strong sink to a source of $CO_2$ on a semi-diurnal time-scale. Seawater $CO_2$ uptake can be underestimated by 73 % and overestimated by 67 %, if these tidal changes are ignored. Awareness of the tidal impacts is required to implement strategic sampling techniques to obtain representative data in these extremely variable - and rarely accessible - systems that play a role in the sensitive balance of the Weddell Sea's net air-sea $CO_2$ exchange.

*Data availability.* Data are available on Pangaea: https://doi.org/10.1594/PANGAEA.946363.

## Appendix A: Sentinel images of Ekström polynya

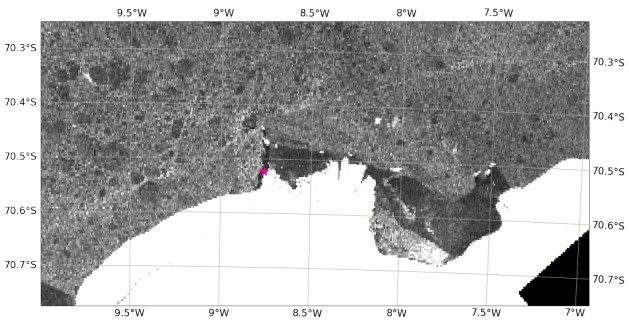

**Figure A2.** Sentinel-1 SAR image of the Ekström Ice Shelf and polynya on 12.01.2015 (PS89), zoomed in (ESA, 2021). The pink star indicates the sampling location for the PS89 tidal observation.

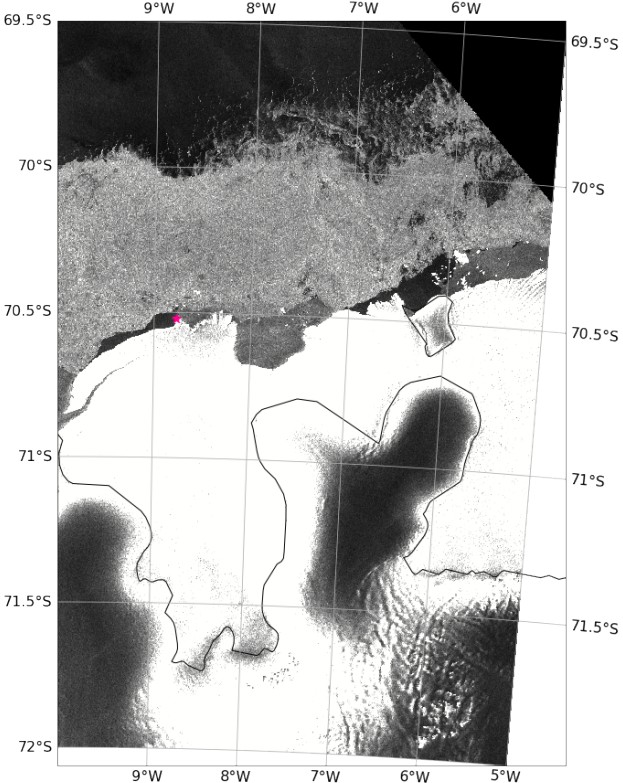

**Figure A3.** Sentinel-1 SAR image of the Ekström Ice Shelf and polynya on 10.01.2019 (PS117) (ESA, 2021). The pink star indicates the sampling location for the PS117 tidal observation.

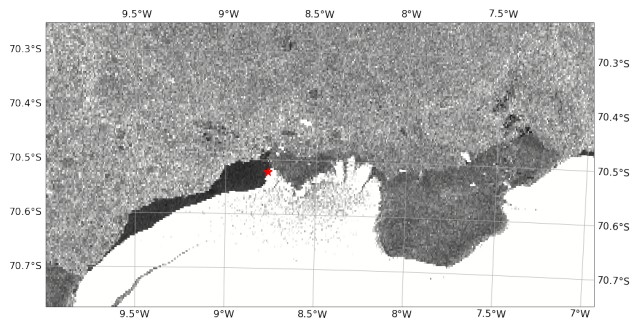

**Figure A4.** Sentinel-1 SAR image of the Ekström Ice Shelf and polynya on 10.01.2019 (PS117), zoomed in (ESA, 2021). The pink star indicates the sampling location for the PS117 tidal observation.

**Table B1.** References to all supporting data and data sources used in this work.

| Data Type | Data Source | Reference |
|---|---|---|
| PS89 and PS117 DIC/TA data | Pangaea | (González-Dávila et al., 2022) |
| PS89 physical oceanographic data AWI CTD (continuous) | Pangaea | (Rohardt and Boebel, 2015a) |
| PS117 physical oceanographic data AWI CTD (continuous) | Pangaea | (Rohardt and Boebel, 2020) |
| PS89 CTD bottle data, incl. nutrient data (discrete) | Pangaea | (Rohardt and Boebel, 2015b) |
| PS117 CTD bottle data, incl. nutrient data (discrete) | Pangaea | (Rohardt et al., 2020) |
| PS89 ADCP | Pangaea | (Witte and Boebel, 2018) |
| PS117 ADCP | Pangaea | (Boebel and Tippenhauer, 2019) |
| PS89 wind speed | Pangaea | (König-Langlo, 2015) |
| PS117 wind speed | Pangaea | (Schmithüsen, 2020) |
| $x$CO$_2$ Syowa | ESRL NOAA | (Dlugokencky et al., 2019) |
| Antarctica boundaries | SCAR ADD | (Gerrish et al., 2020) |
| Ekström CTD data | Pangaea | (Smith et al., 2020b) |
| Sentinel 1 - SAR | European Space Agency | (ESA, 2021) |
| Southern Ocean Bathymetry | IBSCO Version 1.0, Pangaea | (Arndt et al., 2013) |

**Appendix B: Data references**

**Table C1.** CTD sensor precisions.

| Variable | Sensor or method | Expedition | Precision | Reference |
|---|---|---|---|---|
| Conductivity | SBE4c | PS89, manufacturer specification | 0.003 mSc m$^{-1}$ [a] | (Driemel et al., 2017) |
| Salinity | SBE4c | PS117 | 0.0004[b] | (Rohardt and Tippenhauer, 2020) |
| Temperature | SBE3plus | PS89, manufacturer specification | 0.001 °C [a] | (Driemel et al., 2017) |
| Temperature | SBE3plus | PS117 | 0.00000 °C [a] | (Rohardt and Tippenhauer, 2020) |
| Pressure | Digiquartz with TC | PS89, manufacturer specification | 0.015 % full scale | (Driemel et al., 2017) |
| Fluorescence | Wetlabs EcoFLR | PS89 and PS117 | - | (Rohardt and Tippenhauer, 2020) |
| Dissolved O$_2$ | SBE43 | PS89 | 0.43[c] ml L$^{-1}$ | (Boebel, 2015) |
| Dissolved O$_2$ | SBE43 | PS117 | 0.42[d] ml L$^{-1}$ | (Rohardt and Tippenhauer, 2020) |

[a] Average of the residual of post-calibration by Sea-Bird Scientific. [b] Standard deviation of the residual of the difference between the sensor measurements and the Optimare Precision Salinometer samples analysed on board. [c] Standard deviation of the difference in dissolved oxygen Winkler samples analysed on a potentiometric detection system and on a photometric end-point system on board. [d] Standard deviation of the residual of the difference between the sensor measurements and Winkler titration samples.

**Table C2.** Analytical uncertainty of the DIC and TA analyses using the variability of the CRM measurements per CRM batch. n refers to the number of CRMs that were run to obtain the measured values. Values listed for CRMs are certified values by Prof. Andrew Dickson's laboratory at the Scripps Institution of Oceanography of the University of California, San Diego (https://www.ncei.noaa.gov/access/ocean-carbon-acidification-data-system/oceans/Dickson_CRM/batches.html). Net coulometer counts were calibrated against the certified DIC values of the reference material, which was run before and after the sample runs per analysis day. This gave a value for counts per $\mu$mol, for each CRM run. These were averaged and used (along with density and volume) to obtain the concentration of DIC per sample in $\mu$mol kg$^{-1}$.

| Dataset | CRM Batch No. | n | DIC [$\mu$mol kg$^{-1}$] | | TA [$\mu$mol kg$^{-1}$] | |
|---|---|---|---|---|---|---|
| | | | CRM | Measured | CRM | Measured |
| PS89 | 137 | 25 | 2031.90 ± 0.62 | 2031.90 ± 1.22 | 2231.59 ± 0.32 | 2231.71 ± 1.09 |
| PS117 | 176 | 133 | 2024.22 ± 0.82 | 2024.22 ± 3.32 | 2226.38 ± 0.53 | 2226.30 ± 3.18 |
| PS117 | 185 | 17 | 2029.88 ± 0.62 | 2029.88 ± 2.88 | 2220.67 ± 0.58 | 2221.09 ± 0.85 |

# Appendix C: Sampling at study site

**Table C3.** Analytical precisions for nutrient concentrations.

| Dataset | $NO_3^-$ [$\mu$mol kg$^{-1}$] | $PO_4^{3-}$ [$\mu$mol kg$^{-1}$] | $SiO_4$ [$\mu$mol kg$^{-1}$] |
|---|---|---|---|
| PS89[a] | 0.15 | 0.02 | 1.0 |
| PS117[b] | 0.041 | 0.008 | 0.057 |

[a] $\pm$ 1 $\sigma$ of replicate samples for PS89. [b] $\pm$ 1 $\sigma$ of repeat measurements of working standard at similar concentrations as average values for the tidal observation period of PS117: 30, 2, 60 $\mu$mol kg$^{-1}$ for $NO_3^-$, $PO_4^{3-}$, and $SiO_4$, respectively.

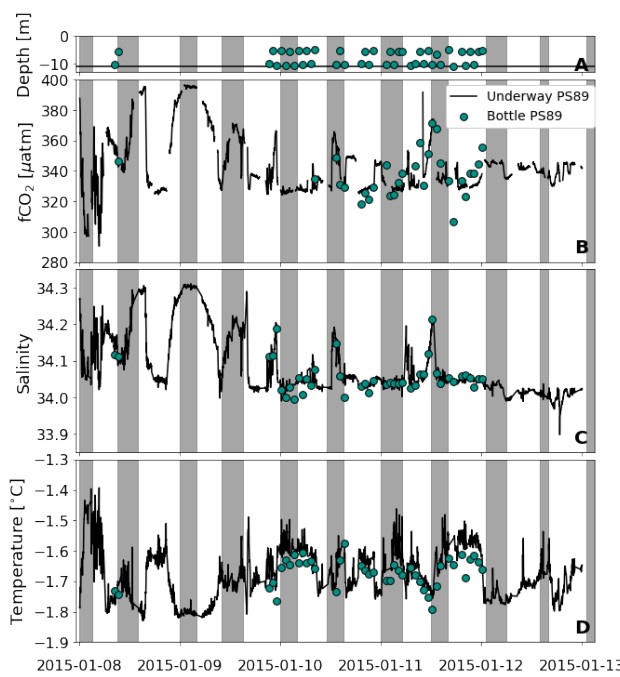

**Figure D1.** A comparison of measurements between the continuous measurements from the on-board underway system and the discrete samples of the surface seawater for PS89. A) Depth at which the surface discrete samples were collected during the tidal observation. The horizontal black line indicates depth at 11 m, which is the depth of the intake for the continuous $f$CO$_2$ measurements. B) The $f$CO$_2$ continuous measurements from the on-board underway system (black line) and the calculated $f$CO$_2$ from the discrete surface samples (circles). C) Same as B), but for salinity. D) Same as B), but for temperature. The grey areas indicate periods of ebbing tide (where the *u* and *v* components of the modelled tidal currents are both positive).

## Appendix D: Underway measurements

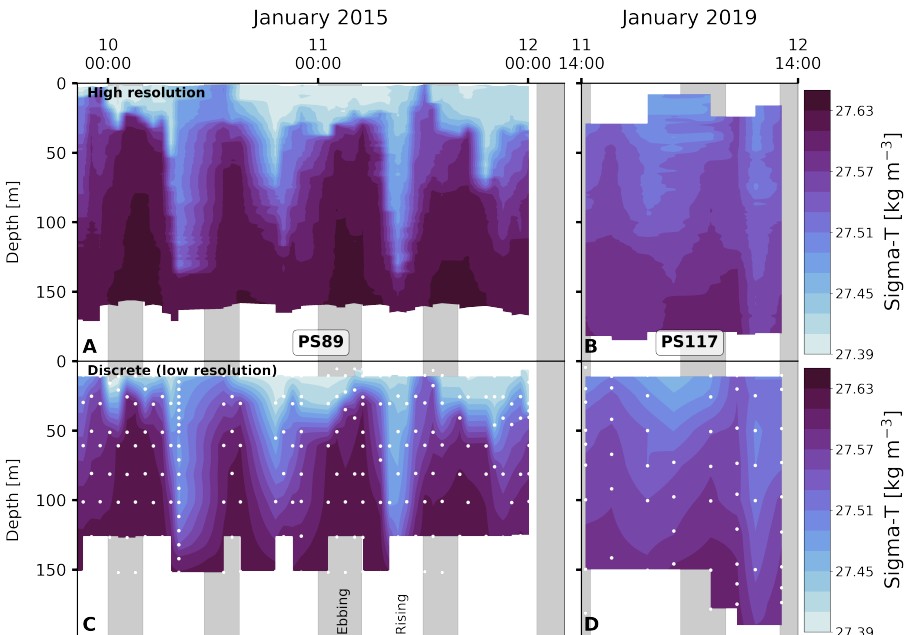

**Figure E1.** A) High vertical resolution profile of sigma-t for PS89. B) High vertical resolution profile of sigma-t for PS117. C) Discrete (low vertical resolution) profile measurements at bottle sampling depth for PS89. D) Discrete (low vertical resolution) profile measurements at bottle sampling depth for PS117. Sampling depths are shown in white markers in C and D.

## Appendix E:  Continuous profile vs. discrete bottle density measurements

To study the tidal cycle, profile measurements from the CTD casts are interpolated over time and depth. The high vertical
resolution profile measurements for salinity and temperature are collected at a 1 dbar resolution on the down-cast, which is
higher than the resolution of the discrete bottle samples and measurements that are collected on the up-cast. Difference in
resolution and natural variability in the water column result in a slight difference between the high vertical resolution profile
(down-cast) and discrete (low vertical resolution) bottle measurements (up-cast) at equal depth. To ensure that the bottle data
with the lower vertical resolution is sufficiently representative, we compare the time and depth-interpolated bottle data to the
high vertical profile measurements interpolated over time for the physical variables. As an example, we show the comparison
for density (Fig. E1). It shows that the resolution of the bottle data sufficiently captures the same features in the water column
as seen in the high resolution profile data. The bottle measurements are therefore reliable for interpretation of the chemical
variables. As the higher vertical resolution profile measurements capture more detail of the features in the water column, these
are used for the time series interpolations wherever applicable.

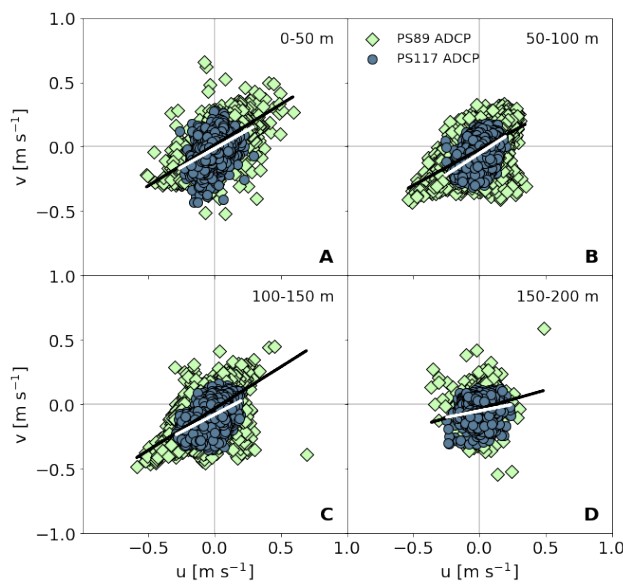

**Figure F1.** Correlations of the $u$ and $v$ components of the ADCP data for the tidal observations during PS89 (green diamonds) and PS117 (blue circles) between 0 - 50 m (A), 50 - 100 m (B), 100 - 150 m (C), and 150 - 200 m (D). All correlations are significant (p-value $< 0.05$) and positive, as shown with the black (PS89) and white (PS117) linear regression lines. A) $v = 0.63 \times u + 0.014$ (PS89); $v = 0.58 \times u$ - 0.01 (PS117). B) $v = 0.56 \times u$ - 0.02 (PS89); $v = 0.66 \times u$ - 0.05 (PS117). C) $v = 0.65 \times u$ - 0.03 (PS89); $v = 0.54 \times u$ - 0.08 (PS117). D) $v = 0.29 \times u$ - 0.034 (PS89); $v = 0.20 \times u$ - 0.05 (PS117).

**Appendix F: Currents**

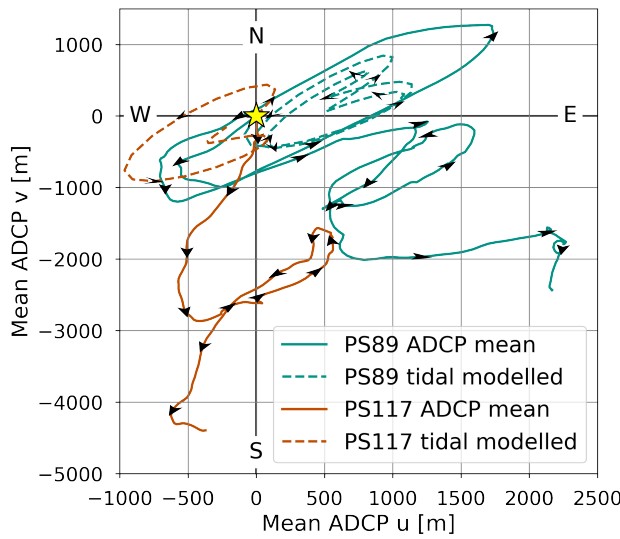

**Figure F2.** The hypothetical distance that a water parcel could have travelled relative to a starting point at the ship's position (indicated by the yellow star) at the starting time of sampling during the PS89 (green) and PS117 (orange) tidal observations. Full lines show the distance travelled using the water column mean ADCP current data. Dashed values show the distance travelled using the modelled water column mean tidal current data. Black arrows indicate the direction of the currents. These calculations assume that the mean currents apply to the larger shelf and polynya region, disregards topographic influences, and ignores the presence of the ice shelf, which is located directly south of the sampling site.

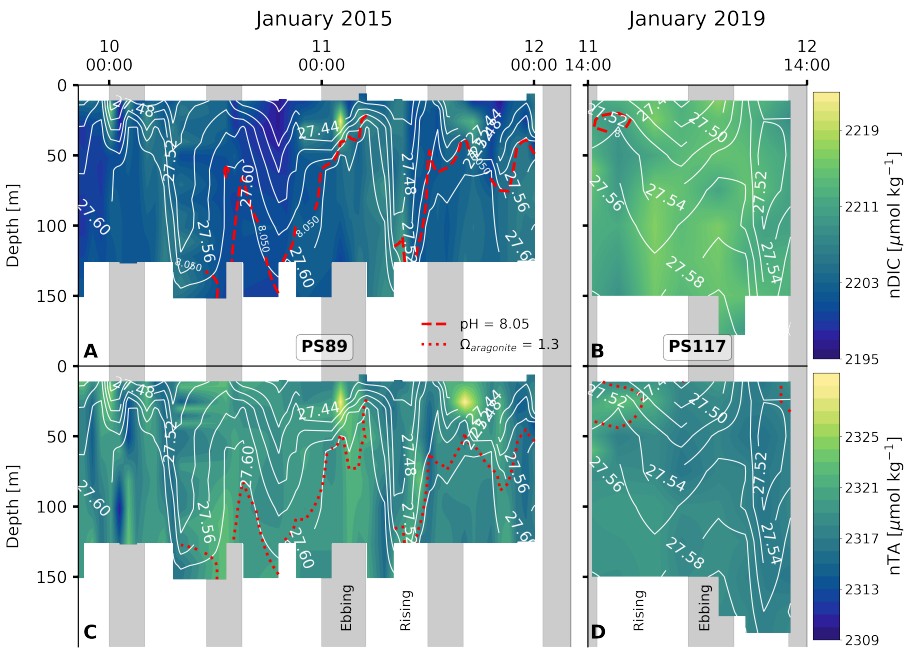

**Figure G1.** Salinity-normalised DIC content at the sampling site during the PS89 (A) and PS117 (B) tidal observations, as well as salinity-normalised TA content for the PS89 (C) and PS117 (D) observations. White contour lines indicate sigma-t in kg m$^{-3}$. Vertical grey shaded areas indicate periods of ebbing tide, as defined in the text. Dashed red lines in A and B indicate the depth at which the pH = 8.05. Dotted red lines in C and D indicate the depth at which the calcite saturation state ($\Omega_{ca}$) = 2.05. These are arbitrary values, used to illustrate the variability in the water column. pH ranged between 8.02-8.12 and 8.02-8.06 for PS89 and PS117, respectively. $\Omega_{ca}$ ranged between 1.95-2.43 and 1.92-2.13 for PS89 and PS117, respectively.

## Appendix G: Tidal variability

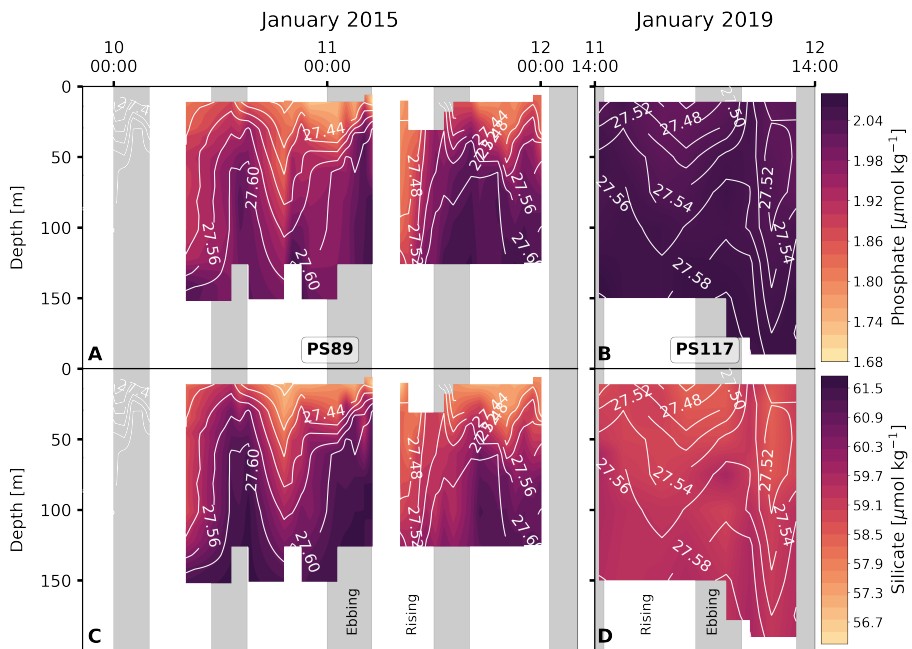

**Figure G2.** As for Fig. G1, but instead for phosphate and silicate content and not including the pH and calcite saturation contours that are shown in Fig. G1.

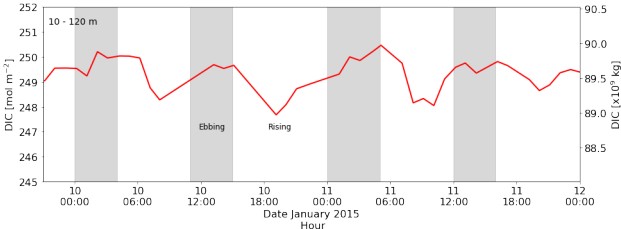

**Figure G3.** Left y axis: total water column DIC content between 10 and 120 m during the PS89 tidal observation. Right y axis: total DIC content between 10 and 120 m for the entire PS89 polynya, using the estimated dimensions in the text and assuming an ellipsoidal area.

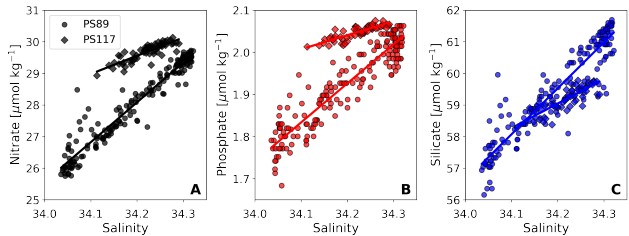

**Figure G4.** Nitrate (A, black), phosphate (B, red), and silicate (C, blue) content plotted against salinity for the PS89 (circles) and PS117 (diamonds) tidal observations, including linear regression lines.

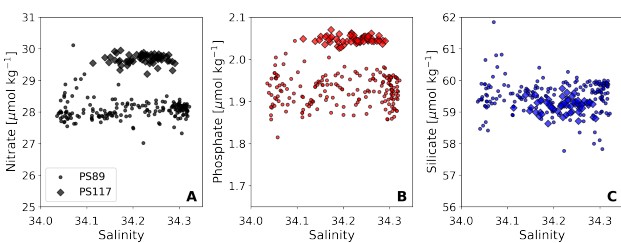

**Figure G5.** Salinity-normalised (following Friis et al. (2003)) nitrate (A, black), phosphate (B, red), and silicate (C, blue) content plotted against salinity for the PS89 (circles) and PS117 (diamonds) tidal observations.

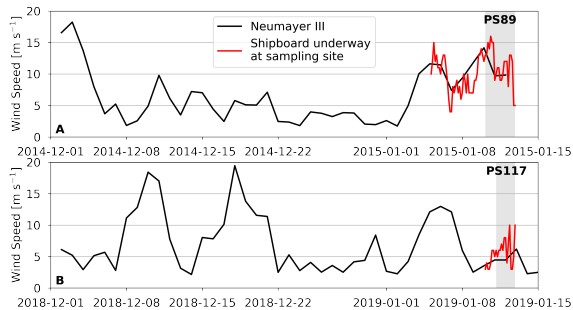

**Figure H1.** A) Wind speed measured at Neumayer III Research Station (black) in the time prior and during the PS89 tidal observation, as well as the ship-board wind speed measurements during the PS89 tidal observation (red). B) Wind speed measured at Neumayer III Research Station in the time prior and during the PS117 tidal observation (black), as well as the ship-board wind speed measurements during the PS117 tidal observation (red). Grey areas indicate the duration of the tidal observations.

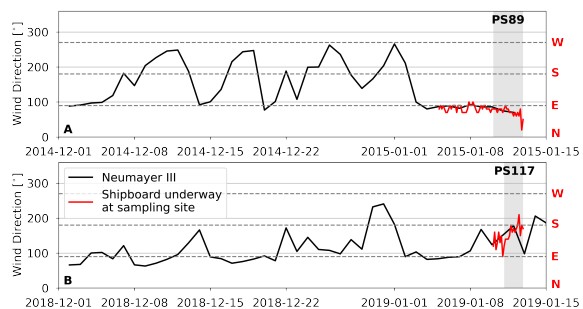

**Figure H2.** A) Wind direction measured at Neumayer III Research Station in the time prior and during the PS89 tidal observation (black), as well as the ship-board wind direction measurements during the PS89 tidal observation (red). B) Wind direction measured at Neumayer III Research Station in the time prior and during the PS117 tidal observation (black), as well as the ship-board wind direction measurements during the PS117 tidal observation (red). Grey areas indicate the duration of the tidal observations. Horizontal dashed lines indicate $90°$, $180°$, and $270°$. North, East, South, and West directions are indicated by the red initials at the right side of the plots.

**Appendix H: Wind**

*Author contributions.* ESD developed the concept for this manuscript, led the writing process, and collected/analysed/processed the PS117 DIC and TA samples. DCEB, MH, GDO, BQ, HJV, MGD, and JMSC provided valuable input and guidance for the development of this work. MGD and JMSC independently collected/analysed/processed the PS89 DIC and TA samples. BQ processed the PS117 ADCP data. DS and SO collected/analysed/processed the PS89 and PS117 nutrient data, respectively. GR ran the tidal model. STK collected and analysed the Winkler $O_2$ samples for the calibration of the dissolved $O_2$ sensors on PS117. All authors contributed to the manuscript.

*Competing interests.* The authors declare that they have no conflict of interest.

*Acknowledgements.* ESD's work is supported by the Natural Environment Research Council (NERC) through the EnvEast Doctoral Training Partnership (NE/L002582/1). Work done by ESD and DCEB for the PS117 expedition was supported by funding from the NERC for the PICCOLO project (NE/P021395/1). Partial support to JMSC and MGD was received from EU FP7 project CARBOCHANGE (Grant agreement No. 264879) for the participation in the PS89 ANT-XXX/2 cruise. MH was partly funded by the European Union's Horizon 2020 Research and Innovation Program under grant agreement N821001 (SO-CHIC). We would like to thank the crew and captains of *R.V. Polarstern* and the scientists for the PS89 and PS117 expeditions for the cooperation on board and the unforgettable experiences. A big thanks to Dr. Janin Schaffer, whose initial description of the tidal influences on the temperature and salinity profiles for the PS117 tidal observation in the PS117 cruise report massively helped the beginning of this work. A special thank you goes out to Salar Karam and Dr. Kirstin Schulz for their patient discussions and valuable insights on physical oceanography of the study site. Finally, we would like to thank our three anonymous reviewers for their valuable comments, which greatly improved the quality of this manuscript.

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
