# Peer review of "The influence of tides on the marine carbonate chemistry of a coastal polynya in the south-eastern Weddell Sea"

_Ocean Science, 2022_

## Referee Comment (RC3)

**Review of**: "The role of tides and sea ice on the carbonate chemistry in a coastal polynya in the south-eastern Weddell Sea" by Droste et al. submitted to *Ocean Sciences*

**Summary**
In this manuscript, the authors present two separate datasets (from January of two different years) documenting the evolution over a few days of physical and biogeochemical parameters measured in a coastal polynya near the Ekström Ice Shelf in the south-eastern Weddell Sea. They show that tides have a significant effect on $fCO_2$ in the polynya surface waters, and by extension on the air-sea flux of $CO_2$, which not only varies considerably, but can also change sign depending on the timing of sampling relative to the tidal cycle. In general, I found this a compelling study and well-written manuscript that succeeding in making its main point – that coastal polynyas near Antarctica are highly dynamic with respect to the carbonate system and that the tidal cycle has a profound influence on whether these features are strong/weak $CO_2$ sources/sinks. I particularly liked the thought exercise that the authors go through in section 4.2 to demonstrate the different answers one might get with regards to air-sea $CO_2$ flux if one did not appropriately account for the influence of the tidal cycle. Below, I have included a number of fairly minor comments and suggestions for the authors to consider, which I hope will help to improve the manuscript.

**Title**: Would it not be more correct to use "The role of tides and sea ice **in** the carbonate chemistry in a coastal polynya in the south-eastern Weddell Sea" or "The **influence** of tides and sea ice on the carbonate chemistry in a coastal polynya in the south-eastern Weddell Sea"? Additionally, the main message of this study seems to be about the tides (and not the sea ice). Of course, sea ice in inherent to the mechanisms the authors invoke to explain their observations, but I found it a little incongruous in the title.

L35-38: These two sentences are somewhat repetitive; can they be combined and made more concise?

L38-39: To what is this sentence referring? Everything outlined in the paragraph is important with respect to $CO_2$ flux estimates, no?

L40-45: I found this paragraph full of general statements that don't really say much – can the authors be more precise, for instance by giving examples or specifics?

L51: "…that has been exchanged with the atmosphere…" – do you mean **acquired from** the atmosphere?

L60: "separate and independent" – I think these two descriptors are redundant.

In general, I think the introduction would be strengthened with some more specific information on what has been shown previously with regards to $CO_2$/carbonate state variables (or even just biogeochemistry and/or hydrography that affects the carbonate system) in coastal polynyas even if the tides have not been considered.

L75-76: Given the subsequent sentence, it doesn't seem particularly informative to state the average area of polynyas.

L89: Sentence seems out of place.

LL92-93: Was the oxygen calibration good? And please can you clarify what you mean in the following sentence by "…malfunctioning on every other cast…"; what are the implications of this?

L99-105: Presumably these measured nutrient concentrations were used in the carbonate system calculations?

Figure 1 caption – where are the bathymetry data from?

Table 1 (and text): For PS117, is eight CTD casts really enough? For PS89, why were two casts conducted on 8 January not included in the case study?

L125: where is Syowa Station?

L135-136: How important might these discrepancies be?

Figure 2 caption – Towards the end of line 2, a space a missing between "The" and "*u*".

L168: Why the choice of a density change of 0.02 kg m$^{-3}$ to define the MLD when the standard is typically 0.03 kg m$^{-3}$ (sensu de Boyer Montégut et al. 2004)?

L174: I don't think the amplitude of variability for salinity can really be considered "striking"

L190 (and elsewhere): The idea that physical processes such as mixing (rather than biogeochemical processes such as photosynthesis, respirations, and/or calcium carbonate formation/dissolution) are primarily responsible for the observed variability in DIC, TA, and fCO$_2$ is alluded to in numerous places in the manuscript. However, the reasoning behind this conclusion is never fully articulated. Since the readership of Ocean Sciences is going to include people not intimately familiar with the drivers of carbonate system variability, I suggest the authors more clearly lay out their logic in this regard, and here may be the place to do it.

L192-194: Can you estimate how much of an effect the increase in atmospheric CO$_2$ might have had? Since the sampling weren't conducted that far apart in time.

L196: Fluorescence is presented here almost as an afterthought, and only for PS117; if these data are going to be shown in the manuscript, the authors should elaborate. I additionally found the text here about silicate to be confusing – why is this meaningful? I feel that if the authors want to draw so much attention to what they consider anomalous or unexpected observations with regard to silicate, they should probably offer an explanation for these observations.

L199: What is meant by "salinity-normalized" nutrient concentrations?

Figure 4 caption – why the choice to highlight pH = 8.05? Similarly, the $\Omega_{aragonite}$ = 1.3; what is the significance of this isoline?

L216: What is meant here by "induced"?

Figure 5 caption – on the second line, remove "or" between "during" and "ebbing"

L227: What was the wind here? In general, I think a more systematic treatment of the winds is required. The values seem to be presented for the first time at random places in the Discussion, which makes it confusing. These values could perhaps be presented in the results?

L232: The idea of a "salinity front" is an important one that crops up throughout the Discussion. However, I don't think it is introduced and explained in a systematic way (unless I missed something). The authors should introduce this idea clearly early on in their Discussion so that its later significance is obvious to the reader.

Figure 6: Is the lack of fluorescence variability during PS117 the result of the colour scale chosen? Also, why are some of the data missing from panel C? Finally, the fluorescence data in panel E seem to me to support the authors' arguments about tidal mixing but they're not discussed at all, nor integrated into the authors' arguments – I think they probably should be.

L240: See above about "salinity front".

L258: And presumably also due to a lack of exchange with the atmosphere?

L260: What do you mean here by "drill sites"? I think this is another example of an idea that needs to be better introduced.

L281-282: Here again, it is not apparent to anyone not deeply familiar with the carbonate system why the nDIC and nTA data suggest that physical processes explain the observed variability. The rationale either needs to be explained here, or explained earlier (see my comment above) and then alluded to here. This is a pretty concept in support of the arguments made in the paper. Additionally, I got to the end of this paragraph as a whole (L293) and wasn't really sure what I should take away from it.

L297: Please explain the significance of the nTA:nDIC ratio.

L299: The word "data" is plural.

L313: What do you mean by "light stability" in this context?

L334: I think this section might benefit from a sentence at the end here summing up what we have learned from the new data presented here rather than simply ending the section after outlining everything we don't know.

L345: "…fluctuates between the bottom (incoming tide) and… (outgoing tide)" – the meaning of this sentence is unclear.

L351: In general, I think it's better to avoid hyperbolic terms such as "drastic".

Figure 7 – I believe that the use of PSU for practical salinity and outdated and such data should be presented with units. Alternately, absolute salinity should be used. Additionally, what is AB? And finally in the 5th line of the caption, should the reference to "single purple markers" actually be to "single white crosses"?

L378: There's some odd repetition here that makes the sentence confusing – please revisit for clarity.

Figure 8 – where is the "dotted line" referred to in the 3rd line of the caption? And if the filled shading indicates the range, does the black line indicate the average wind speed?

L414: What do you mean by "mediating effects"?

L415: What is meant by "repeats are required"? Also, I found the inclusion of "carbonate chemistry" a little incongruous here since that is what is presented in this manuscript. Can you clarify your meaning?

L424: Please see my comments above about the "salinity front".

Data availability: is it acceptable to the journal for the data to only be accessible by contacting the corresponding author? This seems unusual.

Figure A1-A4 – what does the pink star denote? This information should be included in the caption.

Figure G4 – I think the authors should offer a reason for the anomalous silicate data from PS117. Are they certain it's not an analytical error? If not, might it shed some light on the water mass encountered during PS117? I found it a little odd that all these data were presented, the inconsistency highlighted, and then no discussion/explanation offered. Same comment applies to Figure G6.

---

## Author Comment (AC2)

**Authors' Responses to Reviewers' Comments on os-2022-19**

**The role of tides and sea ice on the carbonate chemistry in a coastal polynya in the south-eastern Weddell Sea\**

**Ocean Science July 2022**

Dear anonymous reviewers,

In this document, we address the comments made by anonymous reviewers #1, #2, and #3 on our paper titled, *The role of tides and sea ice on the carbonate chemistry in a coastal polynya in the south-eastern Weddell Sea*\*. Reviewer's comments are **highlighted in blue, are marked bold**, and have been numbered by the authors for easy reference. Some comments have been split up into parts a, b, and c. Responses to these comments are given in black below each comment. All references made to line numbers refer to the line numbers in the originally submitted manuscript (i.e. not the revised manuscript), unless otherwise specified.

An important note: We noticed a mistake had been made in the conversion of atmospheric  $xCO_2$  (ppm) to  $fCO_2$  ( $\mu$ atm). We had used the wrong number and units for the gas constant in the equation by Weiss (1974). The correction of this mistake impacted only the CO2 flux results: the CO2 flux values decreased by ~4 mmol m-2 day-1 for PS89 and ~1 mmol m-2 day-1 for PS117. This correction does not change the overall conclusions of this manuscript, as the variability in CO2 flux remains the same. The only notable change is that the polynya during PS117 is not a clear net source of CO2 anymore. Instead, it is almost neutral. Figures that were adapted:

- Fig. 4: dashed line showing depth of atmospheric fCO2 equivalent corrected.
- Fig. 9: all CO2 flux results corrected.

Numbers in the text were corrected in the abstract, section 4.2, and the conclusion. Please see overleaf for details.

On behalf of all co-authors of this work and myself, I would like to thank the reviewers for their constructive comments, which have greatly improved the quality of this manuscript.

**Elise Droste**

Mario Hoppema, Melchor González-Dávila, Juana Magdalena Santana-Casiano, Bastien Y. Queste, Giorgio Dall'Olmo, Hugh J. Venables, Gerd Rohardt, Sharyn Ossebaar, Daniel Schuller, Sunke Trace-Kleeberg, and Dorothee C. E. Bakker

\*Title has been changed to *The influence of tides on the marine carbonate chemistry of a coastal polynya in the south-eastern Weddell Sea*, according to comment 3.0.

**Adaptations made to the text as a result of the correction of the atmospheric fCO2**

Line numbers are given for the *original* manuscript.

Abstract

• Line 13-14

OG: ... that can range between a strong sink (-20 mmol  $m^{-2} day^{-1}$ ) and a small source (7 mmol  $m^{-2} day^{-1}$ ) on a semi-diurnal timescale.

Revised: ... that can range between a strong sink (-24 mmol m-2 day-1) and a small source (3 mmol m-2 day-1) on a semi-diurnal timescale.

• Line 14-16

OG: If the variability induced by tides is not taken into account, there is a potential risk of overestimating the polynya's  $CO_2$  uptake by 98% or underestimating it by 108% (mistaking it for a source instead of a variable sink), compared to the average flux determined over several days.

Revised: If the variability induced by tides is not taken into account, there is a potential risk of overestimating the polynya's CO2 uptake by 67% or underestimating it by 73%, compared to the average flux determined over several days.

**Section 4.2**

• Line 453-454

OG: The CO2 sink during PS89 is as large as -19.9 mmol  $m^{-2}$  day-1, while the largest CO2 source reaches 6.9 mmol  $m^{-2}$  day-1 (Fig. 8).

Revised: The CO2 sink during PS89 is as large as -23.6 mmol m-2 day-1, while the largest CO2 source reaches 3.1 mmol m-2 day-1 (Fig. 8).

• Line 360-362

OG: Using the fCO2 results based on the discrete water sampling, the average CO2 uptake during the PS89 tidal observation period is  $-8.0 \pm 3.7 \text{ mmol m}^{-2} \text{ day}^{-1} (\pm 1\sigma)$ . For PS117, the average CO2 release to the atmosphere is  $1.4 \pm 0.7 \text{ mmol m}^{-2} \text{ day}^{-1}$ .

Revised: Using the fCO2 results based on the discrete water sampling, the average CO2 uptake during the PS89 tidal observation period is -11.7 ± 3.7 mmol m-2 day-1 (±1 $\sigma$ ). For PS117, the average CO2 release to the atmosphere is -0.1 ± 0.9 mmol m-2 day-1.

• Line 355-360

[revised manuscript text omitted]

Revised: Seawater CO2 uptake can be underestimated by  $\underline{73}$ % and overestimated by  $\underline{67}$ %, if these tidal changes are ignored.

**Anonymous Reviewer #1**

The authors have written an interesting case study that nicely demonstrates how sampling bias can influence marine observations in highly dynamic environments in Antarctic coastal waters. They illustrate this with carbonate chemistry observations from a single location over 1-2 days during two separate years. The authors attribute the observed physical and chemical oceanographic changes to tidally induced currents and mixing.

1.1 Figure 2 shows the expected tidal influence (from a model) alongside the measured currents using an ADCP. Based on this figure alone, it is a little difficult to determine to what extent the tide dominates the observed current movement during the observational period. This is mostly due to the compressed y-axis on panels A, B, G and H. I think the authors have tried to address this with Figure F1, but maybe a plot of the residual u and v component might be more helpful here, or perhaps a progressive vector diagram that shows the trajectory of a water parcel during each period? If tides really are dominant then the water parcel, of course, would pretty much end up back where it started. Although as the authors mentioned in Line 286, the net transport during the experimental period appears to be to the south/southeast. Which would imply a transport path against the prevailing coastal/Weddell Gyre current? Admittedly, this is a minor point. Even if the tidal influence was not as significant, the sampling bias problems that the paper is highlighting would remain unchanged. We thank the reviewer for the suggestions on improving the plot to show the prominence of the tidal currents. We think that Figure F1 does indeed mostly address this comment, but following the suggestion by the reviewer we have now added another figure that shows the hypothetical trajectory of a water parcel starting at the ship's sampling point. This calculation assumes a mean value of the currents for the entire water column and polynya area, disregards topographic effects, and ignores the presence of an iceshelf. It therefore serves as a simple estimate to gain a sense of the strengths/net direction of the currents and the contributions of the tidal currents to the total currents. We decided to place this figure in the appendix, just after Figure F1. We added a reference to the new figure on line 288 of the original manuscript, where we discuss transport.

**1.2 Finally, the caption in Figure D1 incorrectly labels panels C and D.**

We have corrected the caption:

Original text: C) Same as B), but for temperature. D) Same as B), but for salinity. Revised text: C) Same as B), but for salinity. D) Same as B), but for temperature.

**Anonymous Reviewer #2**

This study presents physical and biogeochemical measurements in a polynya and discusses variability and controlling factors during a complete tidal cycle in 2 different years in the eastern Weddell Sea. The data and discussions include using numerical output from a tidal model and considerations of snapshot sampling that may lead to biases and are an important contribution to marine carbonate chemistry, biogeochemical cycling and air-sea CO2 uptake in dynamic environments. The biogeochemical focus is DIC and TA and CO2 fluxes in the context of sea ice and tides.

2.1 Calcium carbonate saturation for both aragonite and calcite are mentioned in the appendix figures but not really in the text. Some additional text in the Introduction and Methods is required to show how these variables were calculated, what they mean for these coastal polynya system and would put the results into greater context with regards the organisms found here.

To address this comment, we have made several changes in the manuscript:

We have renamed the title of section 2.2 from  $CO_2$  flux calculations to Marine carbonate system and  $CO_2$  flux calculations. To this section, we have added the missing information on how the aragonite and calcite saturation were determined.

Added text on line 136 of original manuscript: Alongside fCO2, the PyCO2SYS package simultaneously resolves other carbonate system parameters with DIC, TA, and auxiliary data (listed above) as input parameters (Zeebe & Wolf-Gladrow, 2001; Humphreys et al., 2022a). These include the saturation state for calcite and aragonite (polymorphous forms of calcium carbonate) and pH.

We agree with the reviewer that the relevance of the variability in the polynya system to marine organisms needs to be briefly explained. We decided that the best place in the manuscript to add these lines is between lines 324 and 325 of the original manuscript, because this follows the discussion on the drivers of the variability in the water column and the effect on primary productivity, and precedes the final paragraph(s) in which a summary is given on what we do not know and what we do know based on the results of the current study. We included the following lines in the discussion to link the relevance of the calcium/aragonite saturation variability to the bigger context:

**Added text:**

From an ecological perspective, it is relevant to consider the effect of the carbonate system variability on the diversity, structure, and production of pelagic-benthic organism communities. Calcifying organisms, such as pteropods, foraminifera, and coccolithophores, depend on the seawater calcium carbonate saturation state to form their shells and skeletons, which are made from CaCO3 (Orr et al., 2005). The pH in the Ekström polynya varied between 8.02-8.12 and 8.02-8.06 for PS89 and PS117, respectively (see Fig. G1 for vertical variability). The saturation state of aragonite ( $\Omega_{ar}$ ; the less stable polymorph compared to the other common CaCO3 polymorph: calcite) concurrently varied between 1.22-1.52 and 1.21-1.34, respectively. A contour in Fig. 4C and D at an arbitrary value of 1.3 for  $\Omega_{ar}$  gives a sense of the vertical variability driven by tides. Even at the lowest pH values recorded here, the  $\Omega_{ar}$  does not fall under 1, which means that the marine chemical environment does not thermodynamically promote CaCO3 dissolution. The dynamic nature of the polynya might foster a

resilience among the pelagic and benthic organism communities to rapid (semi-diurnal) changes of  $\Omega_{ar}$ . However, even at a carbonate saturation level > 1, the rate of biogenic calcification rate has been shown to be affected by the CaCO3 saturation state (Feely et al., 2004). The Southern Ocean is especially vulnerable to ocean acidification driven by marine anthropogenic CO2 uptake (Orr et al., 2005; Negrete-García et al., 2019). Tidally induced variability may increase the sensitivity of high-latitude coastal systems to shoaling aragonite and calcite saturation state horizons. Alongside ecological impacts of tides along the Antarctic coastline, future studies can look into the tidal impacts on long-term changes of the vulnerability of pelagic-benthic organism communities.

While making these alterations in the text, we noticed that there was a mistake in the caption for Figure 4, which has now been corrected:

Original text in figure 4 caption: The red dotted line represents the depth at which the seawater pH = 8.05.

Revised text in figure 4 caption: The red dotted line represents the depth at which the  $\Omega_{ar} =$  1.3.  $\Omega_{ar}$  is lower at depths below this line. The value 1.3 is arbitrary to illustrate the vertical variability in the water column.  $\Omega_{ar}$  ranges between 1.22-1.52 and 1.21-1.34 for PS89 and PS117, respectively.

As the contour for aragonite is already shown in Figure 4, we have changed the contour in panels C and D of figure G1 from aragonite saturation state = 1.3 to calcite saturation state = 2.05. The corresponding text in the caption has been edited accordingly.

**2.2 It would also be helpful to include more discussion of and reference to the theoretical lines drawn in Figure 5, whereby a short description of key processes that drive variability in the carbonate system in the Introduction would improve the understanding.**

We thank the reviewer for this comment, which has helped us restructure and edit a paragraph in section 4.1 and has resulted in a much clearer link between our arguments and how Fig. 5 supports them. We think this is a suitable place to refer to the theoretical lines (instead of including an explanation in the introduction, as suggested), because this is where we discuss the biogeochemical processes and we can directly link it to our findings (allowing us to be concrete and avoiding repetition). This part of the text has been revised in combination to our response to comment 3.31. Please see page 15 of the revised manuscript to see the changes we made to the text.

**2.3 Figure 3 green markers in panel A are difficult to see.**

We have increased the size of the markers and also the width of the markers' edges to make them clearer.

**2.4 Figure D1 panels C and D descriptions are reversed in the caption.** We have corrected this mistake. Please see our response to comment 1.2 by reviewer #1.

**2.5 Figure E1 interpretation would be assisted by marking depths of discrete samples in panels C and D to better compare to higher vertical resolution in panels A and B.** Done. We have also added this information in the caption of Fig. E1:

Added text to caption Fig. E1: Sampling depths are shown in white markers in C and D.

2.6 There is assumption that the interpretation of biogeochemical data from the discrete samples is reliable as the physical variables from the high resolution CTD data, however additional processes such as primary production/respiration, location of a deep Chl maximum... would imprint additional variability particularly in the surface layer that is not captured by changing salinity and temperature (water mass) interactions. A comment in the text to consider this and consider adding references to support the statement that would complement the discussion.

We agree to include a comment on the consideration of missing variability due to the vertical sampling resolution. We decided that a relevant spot in the manuscript to do so is on line 303, which is the end of the paragraph in which we discuss the variability in the nDIC and nTA content and which processes might have affected those values. We have not included a reference here, but instead have provided more clarification (supported by references) on the role of physical processes and the relation to TA and DIC elsewhere in the text. Please see our responses to comments 3.18 and 3.31.

Line 303: Finally, even though there is good agreement between the high vertical resolution sensor data and the discrete bottle data for salinity and temperature (exemplified with density in Fig. E1), biogeochemical processes could imprint additional variability in the DIC and TA profiles that are not reflected in salinity and temperature measurements. We must therefore consider that the discrete seawater samples might not have captured the full scale of the variability in the polynya, limiting our interpretation of relevant biogeochemical processes.

2.7 Figure G1 determining the difference between red dashed and dotted lines was difficult, perhaps a more striking difference would assist here (e.g. different colours). Aragonite saturation is mentioned here and would benefit from an introduction in the main text in terms of the definition and how it is determined, and relevance of the value depicted by the red line here, low value towards 1 relevant for calcifying organisms? To address this comment, we tried pairing a number of different colour combinations, while making sure that the choice of colours remains colour-blind friendly. Red was the colour that contrasted best with the colourmap for both colour-blind and non-colour-blind people. We have changed the colour of the dotted line to cyan to make the difference clearer with the dashed line.

We have added a section to the methods to explain how aragonite and calcite saturation states were determined and we have added text to the discussion to explain the relevance. Please also see our response to comment 2.1.

**2.8** Figure G2 calcite saturation is shown here, check consistency with Figure G1 and include definitions and how they are determined in the text.**

This comment by the reviewer helped us discover a slight mistake that we made in Figure G1, although there might have been some confusion as to what is shown in G2. Figure G1 was supposed to show the contours of calcite saturation state = 2.05 (bottom panels), but instead still had the saturation state for aragonite = 1.3. This has been corrected (see response to comment 2.1). We have also changed the caption of Figure G2 to make it clearer that it has a similar set up as for G1, but showing phosphate and silicate content instead of nDIC and nTA and does not show the pH and calcite saturation contours:

Original caption Figure G2: As for G1, for phosphate and silicate (excluding the pH and calcite saturation contours).

Revised caption Figure G2: As for **Fig.** G1, **but instead for phosphate and silicate content and not including the pH and calcite saturation contours that are shown in Fig. G1**.

**Anonymous Reviewer #3**

In this manuscript, the authors present two separate datasets (from January of two different years) documenting the evolution over a few days of physical and biogeochemical parameters measured in a coastal polynya near the Ekström Ice Shelf in the south-eastern Weddell Sea. They show that tides have a significant effect on fCO2 in the polynya surface waters, and by extension on the air-sea flux of CO2, which not only varies considerably, but can also change sign depending on the timing of sampling relative to the tidal cycle. In general, I found this a compelling study and well-written manuscript that succeeding in making its main point – that coastal polynyas near Antarctica are highly dynamic with respect to the carbonate system and that the tidal cycle has a profound influence on whether these features are strong/weak CO2 sources/sinks. I particularly liked the thought exercise that the authors go through in section 4.2 to demonstrate the different answers one might get with regards to air-sea CO2 flux if one did not appropriately account for the influence of the tidal cycle. Below, I have included a number of fairly minor comments and suggestions for the authors to consider, which I hope will help to improve the manuscript.

**3.0** Title: Would it not be more correct to use "The role of tides and sea ice in the carbonate chemistry in a coastal polynya in the south-eastern Weddell Sea" or "The influence of tides and sea ice on the carbonate chemistry in a coastal polynya in the south-eastern Weddell Sea"? Additionally, the main message of this study seems to be about the tides (and not the sea ice). Of course, sea ice in inherent to the mechanisms the authors invoke to explain their observations, but I found it a little incongruous in the title.

We thank the reviewer for the correction and suggestions. The title has been changed to: The **influence** of tides **on** the **marine** carbonate chemistry **of** a coastal polynya in the southeastern Weddell Sea. Note, we have omitted "and sea ice" from the title, as we agree with the reviewer that it is a little incongruous.

**3.1 L35-38: These two sentences are somewhat repetitive; can they be combined and made more concise?**

We have revised these two sentences as follows:

Original text: The prolonged ice-free conditions allow direct gas exchange to occur, a longer time window for equilibration of  $CO_2$  with the atmosphere, as well as prolonged biological productivity (Hoppema and Anderson, 2007). Substantial biological activity is observed in coastal polynyas around the Antarctic continent (Arrigo and van Dijken, 2003).

Revised text: The prolonged ice-free conditions allow direct gas exchange to occur **over a time window that is potentially long enough for** equilibration of CO2 with the **atmosphere** (**Hoppema** and Anderson, 2007). Additionally, substantial biological activity (**consuming CO2**) is observed in coastal polynyas around the Antarctic continent (Arrigo and van Dijken, 2003).

**3.2 L38-39:** To what is this sentence referring? Everything outlined in the paragraph is important with respect to CO2 flux estimates, no?**

The aim of this sentence was to conclude the paragraph with the message that coastal polynyas are important to consider in  $CO_2$  flux estimates of polar regions. However, we can see the confusion caused by the way the sentence is phrased. We have thus clarified this it in the following way:

Original text: This makes them important with regard to air-sea CO2 flux estimates.

Revised text: Their prolonged ice-free conditions and biological productivity mean that it is important to accurately represent the air-sea CO2 flux of coastal polynyas.

**3.3 L40-45: I found this paragraph full of general statements that don't really say much** – can the authors be more precise, for instance by giving examples or specifics? We have incorporated this comment into our response to comment 3.6.

**3.4 L51: "...that has been exchanged with the atmosphere..." – do you mean acquired from the atmosphere?**

We did not mean "acquired", as natural CO2 may be released from the ocean to the atmosphere as well. This comment prompted us to change the sentence slightly in the following manner (which we think is indeed clearer):

Original text: "... along with any  $CO_2$  that has been exchanged with the atmosphere at the continental shelf."

Revised text: ... along with any **modifications to the CO2 content through gas exchange** with the atmosphere at the continental shelf.

**3.5 L60: "separate and independent" – I think these two descriptors are redundant.** We agree and have omitted "separate and independent" from the sentence.

Original text: We will present biogeochemical observations for two separate and independent tidal sampling campaigns ...

Revised text: We will present biogeochemical observations **for two** tidal sampling campaigns ...

**3.6** In general, I think the introduction would be strengthened with some more specific information on what has been shown previously with regards to CO2/carbonate state variables (or even just biogeochemistry and/or hydrography that affects the carbonate system) in coastal polynyas even if the tides have not been considered.

We have adapted the introduction to incorporate this helpful and constructive feedback. It required some restructuring of the introduction to maintain a good flow of the text. We have added examples with additional references and replaced sentences that were rather vague with more concrete ones. Please see the revised introduction section in the revised version of the manuscript.

**3.7 L75-76:** Given the subsequent sentence, it doesn't seem particularly informative to state the average area of polynyas.**

We agree that this sentence can indeed be removed (and we indeed did so).

Original sentence removed: Coastal polynyas in the eastern Weddell Sea have an average area of 7.75 x  $103 \text{ km}^2$  in the summer and  $1.12 \times 103 \text{ km}^2$  in winter (coastal polynyas numbers 12-17 in Arrigo and van Dijken (2003)).

However, we think it might be useful to the reader to put the size of the polynya in the case studies into perspective relative to other polynyas in the region. This why we have slightly modified the sentence on line 81-82 of the original manuscript in the following manner:

Original sentence: Generally, the polynya during both tidal observations at this location is considered to be relatively small.

Revised sentence: Given that the average area of coastal polynyas in the eastern Weddell Sea in summer is  $7.75 \times 10^3 \text{ km}^2$  (coastal polynyas numbers 12-17 in Arrigo and van Dijken (2003)), the polynya during both tidal observations at this location is considered to be relatively small.

We think that in the revised text, the selected information from the omitted sentence is more useful to the reader.

**3.8 L89: Sentence seems out of place.**

The purpose of this sentence was to make clear to the reader where the Ekström Ice Shelf and the sea ice were relative to the polynya/sampling site, as this is important to understand for the interpretation of the results (for example, it is important to know that there was no sea ice between the polynya and the ice shelf). We agree with the reviewer that its placement in the text does break the flow a bit. We have therefore moved the sentence on line 89 to line 74 in the original manuscript:

Moved sentence: The polynya is bordered by the Ekström Ice Shelf along its southern edge, and by sea ice along the rest of its perimeter.

We think that this is a better place in the text for this information, as this paragraph already contains information about the location of the polynya.

**3.9 LL92-93:** Was the oxygen calibration good? And please can you clarify what you mean in the following sentence by "...malfunctioning on every other cast..."; what are the implications of this?**

Yes, the oxygen calibration was good (see precisions in Table C1). To clarify: The CTD casts were alternated between the CTD from AWI and an Ultra Clean CTD (UCC) from NIOZ. The oxygen optode on UCC from NIOZ malfunctioned, which is why we cannot include the oxygen data from these 4 casts. The oxygen data on the AWI CTD was fine. We thought an explanation of the two different CTD systems was perhaps a little irrelevant to include in the manuscript (as most of their system is the same between them), which is why we wrote that the optode malfunctioned during "every other cast".

The implication of this is that, for dissolved oxygen and fluorescence, we only have half the number of data for the PS117 tidal observation. This means that the resolution in time is lower and we would miss out on some of the variability in the water column through time. However, we decided to still show the water column data for oxygen and fluorescence, because the lack of variability in other variables, such as nutrients, DIC/TA, and salinity, indicate that there is not a lot of variability in the oxygen data that we could be missing out on due to the missing oxygen profiles from the UCC.

The same counts for fluorescence, for which we only have data from the AWI CTD. We have now included this in the text (see below).

We recognise that it is necessary to elaborate on this topic. We have therefore revised the lines highlighted by the reviewer to clarify the use of different rosettes (on one of which the optode malfunctioned), and refer to added text in the results section in which we explain the minor implication of this.

Original text: The oxygen optode sensor was malfunctioning on every other cast during the PS117 tidal observation period and thus its data had to be excluded from further analysis.

Revised text: During the PS117 tidal observation period, two different CTD rosettes with their own set of sensors were used in alternation. On one of these two CTD rosettes, the oxygen optode sensor malfunctioned and thus its data had to be excluded from further analysis. The fluorescence data for this particular rosette is also not available. The minor implication of this is addressed in Section 3.3.

Added text on line 202 of the original manuscript (at the end of section 3.3):

Due to complications with the data for dissolved oxygen and fluorescence on four of the CTD casts of the PS117 tidal observation, these data were excluded from analysis. The implication of this reduced temporal resolution is that we risk losing representation of water column variability in our dataset for these two variables. However, the low water column variability over time for the physical and other biogeochemical variables (for which we do have data from every cast) strongly suggest that there likely is not a lot of variability in the dissolved oxygen or fluorescence content that we are missing out on due to missing profiles.

**3.10 L99-105: Presumably these measured nutrient concentrations were used in the carbonate system calculations?**

The measured nutrient concentrations were indeed used in the carbonate system calculations. We have now included this in the text in Section 2.2 (lines 129-130 in the original manuscript), where we explain how the seawater fCO2 was determined.

Original text: The fCO2 of surface seawater (fCO2sw) is determined from the DIC and TA concentrations of the shallowest discrete seawater values ...

Revised text: The fCO2 of surface seawater (fCO2sw) is determined from **the DIC**, **TA**, **nitrate**, **and phosphate content**, **as well as the temperature and salinity**, of the shallowest discrete seawater samples ...

**3.11 Figure 1 caption – where are the bathymetry data from?**

The bathymetry is from IBSCO Version 1.0. This information was supposed to be included in Table B1, but we now noticed that it was missing. We thank the reviewer for pointing this out. We have included the source of this data in Table B1 and also in the caption of Figure 1:

Original text: Weddell Sea with bathymetry.

Revised text: Weddell Sea with bathymetry: IBSCO Version 1.0 (Arndt et al., 2013).

**3.12** Table 1 (and text): For PS117, is eight CTD casts really enough? For PS89, why were two casts conducted on 8 January not included in the case study?

For PS117: A CTD cast was done every ~3 hours and during times of both ebbing and rising tide. We believe that this is indeed enough to capture a representative variability of the water column during this (very short) time period. We have included the following line in the caption for Table 1 to clarify this:

**Added text: For both observation periods, casts were lowered into the water during times of ebbing and rising tide.**

For PS89: The two casts from 8 January were not included in the case study, because they were too far removed in time from the other observations to be able to interpolate between them. We decided that including them would not substantially contribute to the message we aim to discuss. However, we have added a note to the footer of table 1, and have now also included the CO2 flux based on the discrete surface seawater sample on the 8th of January in Fig. 8, because the underway data gives context to that "lonely" datapoint.

**3.13 L125: where is Syowa Station?**

Syowa Station is a Japanese permanent station on the Antarctic coastline at  $69^{\circ}$ S latitude. We used their atmospheric xCO2 time series data, because it is relatively close (in terms of stations that maintain a time series) to our study site. We compared the time series to the xCO2 time series from the South Pole, but there was no consistent difference between the two datasets and insignificant difference between the two, especially for the monthly averages. This indicates that the atmosphere is very well mixed for CO2 on Antarctica and along the coastlines. We therefore opted to use the data from Syowa Station, also because it is at the coast and closer in proximity to our study site.

We have included the coordinates for Syowa Station when it is first mention in the text (line 125 in original manuscript).

**3.14 L135-136: How important might these discrepancies be?**

Discrepancies between the measured and calculated  $fCO_2$  values are important to consider (and to resolve/understand) when deciding which data to use to accurately determine air-sea  $CO_2$  flux calculations. The main purpose of our study is to illustrate the relative variability of the system (and therefore also the  $CO_2$  flux), and therefore *small* discrepancies would not discredit the messages that we are trying to convey. However, as can be seen in Fig. 8, the  $CO_2$  flux results based on the measured and calculated  $fCO_2$  values actually agree very well, despite the small discrepancies. We therefore consider them not to be very important. To make this clearer to future readers of the manuscript, we have added the following text on line 136 of the original manuscript:

Added text line 136: As is shown in Section 4.2, these discrepancies are not large enough to affect the agreement between the CO2 flux estimates based on these two sets of fCO2 data.

**3.15 Figure 2 caption** – Towards the end of line 2, a space a missing between "The" and "u".

Corrected.

**3.16 L168:** Why the choice of a density change of 0.02 kg m-3 to define the MLD when the standard is typically 0.03 kg m-3 (sensu de Boyer Montégut et al. 2004)?

We calculated the mixed layer depth using a range of density differences to test the robustness of the final result for the MLD. The results for MLD did not differ substantially when using density differences between 0.01-0.05 kg m-3. We were not aware that 0.03 kg m-3 is considered a standard, and because a density of 0.02 kg m-3 seemed to capture the changes in MLD sufficiently well, we decided to use this. We agree that it is better to stick to more commonly used methods, and so we have now recalculated the mixed layer depths and updated the markers in Fig. 3. The MLD results (and conclusions based on them) did not change.

0.02 kg m-3 has been changed to **0.03** kg m-3 on line 167 of the original manuscript and in the caption of Fig. 3.

**3.17 L174: I don't think the amplitude of variability for salinity can really be considered "striking"**

Replaced "striking" with "much".

Original text: ... the amplitude of variability for salinity is strikingly smaller compared to PS89.

Revised text: ... the amplitude of variability for salinity is **much** smaller compared to PS89.

3.18 L190 (and elsewhere): The idea that physical processes such as mixing (rather than biogeochemical processes such as photosynthesis, respirations, and/or calcium carbonate formation/dissolution) are primarily responsible for the observed variability in DIC, TA, and fCO2 is alluded to in numerous places in the manuscript. However, the reasoning behind this conclusion is never fully articulated. Since the readership of Ocean Sciences is going to include people not intimately familiar with the drivers of carbonate system variability, I suggest the authors more clearly lay out their logic in this regard, and here may be the place to do it.

We agree with the reviewer that this is a concept that could have been better introduced. We decided to include it in line 189 of the original manuscript. We allude back to it in lines 281-282 (see our response to comment 3.31).

Original text line 189-191: Salinity-normalised DIC (nDIC) and TA (nTA) profiles (according to Friis et al., 2003) lose much of the semi-diurnal variability, suggesting that physical processes, such as advection and mixing, are the dominant drivers of the observed variability (Fig. G1).

**Revised text:**

In the ocean, changes to salinity are driven by oceanographic processes, such as dilution, ice formation, and mixing. These physical processes also impact DIC and TA, which is why DIC and TA are often strongly correlated to salinity (Middelburg et al., 2020). This is also the case in our dataset. However, DIC and TA content are additionally a function of biological (e.g. photosynthesis, respiration, and remineralisation) and chemical (e.g. CaCO3 dissolution and precipitation) processes (Zeebe & von Wolf-Gladrow et al., 2001). To be able to study the role of biogeochemical processes on DIC and TA content, it is useful to separate the effect of physical processes, such as dilution and mixing of different water masses, from the rest. This can be done by normalising the DIC and TA values to the salinity, which we have done here according to methods by Friis et al. (2003). The salinity-normalised DIC (nDIC) and TA

(nTA) **profiles lose** much of the semi-diurnal variability seen in the profiles **of the non-salinity-normalised values**, suggesting that physical processes are the dominant drivers of the observed variability **in DIC and TA** (Fig. G1).

**3.19 L192-194:** Can you estimate how much of an effect the increase in atmospheric CO2 might have had? Since the sampling weren't conducted that far apart in time.**

To be able to estimate the contribution of atmospheric  $CO_2$  to the DIC of seawater, we would need to know how much the water has equilibrated with the atmosphere. Given that a) this area is mostly sea ice covered during the year, b) the area and duration of the polynya is variable in the summer, and c) our current work has shown that it is a very dynamic system, we cannot estimate the degree of equilibration.

However, we can make a rough estimation for the upper bound to the contribution of atmospheric CO2 increase to the increase in DIC by assuming equilibration and assuming all other variables (TA, salinity, temperature, nutrients) remain the same. The increase in atmospheric fCO2 between January 2015 (377  $\mu$ atm) and January 2019 (387  $\mu$ atm) is 10  $\mu$ atm.

We used the average values of the upper 10 m during PS89 for salinity, temperature, TA, seawater fCO2 (originally calculated from TA and DIC), and nutrients to calculate the surface DIC content. We then repeated the calculation, but added 10  $\mu$ atm to the average seawater fCO2 (to simulate equilibration), and subtracted the former from the latter. The result is ~6  $\mu$ mol kg-1 increase in DIC.

We have added this upper-bound estimate to the manuscript:

Line 192-194 of original manuscript: A part of this increase in nDIC over time could be result of the increase in atmospheric  $CO_2$ , assuming at least partial equilibration with the atmosphere.

Revised version: A part of this increase in nDIC over time could be result of the increase in atmospheric CO2, assuming at least partial equilibration with the atmosphere. The atmospheric fCO2 increase (10  $\mu$ atm) alone could contribute ~6  $\mu$ mol kg-1 to the surface DIC content if all other variables remained the same. This upper-bound estimate is based on average values of the top 10 m during PS89 and assumes equilibration of the surface water with the atmosphere.

**3.20** L196: **a**) Fluorescence is presented here almost as an afterthought, and only for PS117; if these data are going to be shown in the manuscript, the authors should elaborate. **b**) I additionally found the text here about silicate to be confusing – why is this meaningful? I feel that if the authors want to draw so much attention to what they consider anomalous or unexpected observations with regard to silicate, they should probably offer an explanation for these observations.

a) We agree that more attention could have been given to the fluorescence. We decided not to elaborate too much on fluorescence in the Results section, because we have now revised parts of the text in the Discussion section to better incorporate the relevance of the fluorescence data into the support of our arguments. Please see the third part of our response to comment 3.27 for the details. We have, however, revised line 196 to at least compare the fluorescence

during PS117 explicitly to that during PS89. Note that this revision is merged with revisions made in response to part b) (3.20 b).

Original text line 196: Fluorescence was barely detected during PS117 (Fig. 6).

**Revised text line 196: Fluorescence is here used as a proxy for the presence of photosynthetic cells. While the rising tide increased fluorescence in the water column during PS89, it was barely detected during PS117 (Fig. 6).**

b) Note: While addressing this comment (3.20 b), we are also addressing comment 3.46, because we feel that both comments are on the same topic. We agree with the reviewer that more needs to be said about the nutrient (incl. silicate) concentrations and how they contribute to our arguments. We have double checked the silicate data and have found no analytical errors. We have confidence in the values that we have presented. Silicate does not behave the same way as nitrate or phosphate, as its concentrations are mainly altered by diatom growth (incorporated into opal) or remineralisation, whereas nitrate and phosphate are strongly affected by photosynthesis and respiration. Silicate remineralisation happens only in the deep waters and in sediments. We can thus expect a different relationship for phosphate and nitrate, vs. silicate. These relationships may be different in different years.

While we are unable to fully explain the nutrient content differences between the two case studies (because we do not have all the necessary biological measurements to be able to do so) we have added a small discussion.

Our revisions to the text in response to this comment are merged to the revisions made in response to part a) (3.20 a).

**Revised text on page 10 of the *revised* manuscript:**

PS117 nitrate (28.9 - 30.1 umol kg-1), phosphate (2.0 - 2.1 215 umol kg-1), and dissolved oxygen (322.0 - 333.2 umol kg-1) concentrations throughout the water column have similar values as those at the bottom of the water column during PS89 (Fig. 6, G2, G4). This is not the case for silicate, for which its PS117 values lie around the mean silicate values measured for PS89 (60.0 ±1.1 umol kg-1; Fig. G4). This observation illustrates that silicate behaves differently than phosphate and nitrate, as its content is primarily affected by other processes, i.e. by diatom growth and remineralisation at depth or in the sediment rather than by photosynthesis and biological respiration (Sarmiento, 2013). Similarly to nDIC and nTA, the nutrients were salinity-normalised (following Friis et al., 2003). Consistent to nDIC and nTA, no obvious deviations from the mean salinity-normalised values are observed that could indicate a dominant biological influence. It supports the observation made above that the dominant driver of the variability observed *within* each tidal case study is mostly physical. For the salinitynormalised silicate content, the averages of both tidal observations are similar to each other  $(59.5 \pm 0.5 \text{ umol kg}^{-1} \text{ for PS89 and } 59.2 \pm 0.2 \text{ umol kg}^{-1} \text{ for PS117; Fig. G5}),$ indicating that processes affecting silicate content did not differ much between the two case studies. However, a consistent offset between the case studies is observed for the salinity-normalised values for nitrate and phosphate, where PS117 values are on average higher by 1.6 umol kg-1 and 0.1 umol kg-1, respectively. Therefore, even though biogeochemical processes might not be able to explain the variability *within* each tidal observation, these results suggest that the polynya may have had a higher input and/or a lower loss of nitrate and phosphate during January 2019 compared to January 2015.

Fluorescence is here used as a proxy for the presence of photosynthetic cells. While the rising tide increased fluorescence in the water column during PS89, it was barely detected during PS117 (Fig. 6). In the absence of active cells that can photosynthesise/respire, remineralisation enhancing the nitrate and phosphate content may have been an important process in the water observed during PS117.

We then allude to the higher salinity normalised values for nitrate, phosphate, and DIC, on page 16 of the *revised* manuscript, where we build our argument that PS117 had more influence from the ice shelf cavity water. This required us to shuffle a few lines (i.e. lines 275-280 of original manuscript), to avoid unnecessary repetition and maintain a good flow of the text. For the same reasons, we have started a new paragraph (at line 316 of original manuscript) to improve the structure of the text.

**3.21 L199: What is meant by "salinity-normalized" nutrient concentrations?**

We mean that the nutrients have been normalised to salinity using the linear regression method by Friis et al. (2003), similarly to how we normalised DIC and TA to salinity. As a response to comment 3.18, we have introduced the reasoning behind normalising DIC and TA to salinity. We think that this, in combination with the following minor change to the text, has improved the clarity that this comment addresses.

Original text: Whereas the salinity-normalised concentrations for nitrate and phosphate during PS117 have an average that is significantly higher ...

Revised text: Whereas the salinity-normalised concentrations for nitrate and **phosphate** (following Friis et al., 2003, similarly to nDIC and nTA) during PS117 have an average that is significantly higher ...

**3.22 Figure 4 caption – why the choice to highlight pH = 8.05? Similarly, the $\Omega$ aragonite = 1.3; what is the significance of this isoline?**

We chose pH = 8.05 and  $\Omega$ aragonite = 1.3 arbitrarily and only to give a sense of the vertical variability in the water column over time. We have added text at lines 324 and 325 of the original manuscript to better introduce and explain the relevance of pH and  $\Omega$ aragonite (shown in Figure 4) and pH and  $\Omega$ calcite (shown in in figure G1). Please see our response to comment 2.1. The added lines, as well as the captions of Figures 4 and G1, also more clearly state that these values are essentially arbitrary but serve the purpose to illustrate how the depth of these horizons can vary. The range of values for pH,  $\Omega$ aragonite,  $\Omega$ calcite are also given in the respective captions.

**3.23 L216: What is meant here by "induced"?**

The line was missing a part of the sentence, probably got lost in the reiterations of the revisions of the manuscript. We have now completed this sentence:

Original text: induced by intrusions of warmer water, such as mWDW and AASW, underneath the ice shelf ...

Revised text: **formed by melting of glaciers, and of ice shelves** induced by intrusions of warmer water, such as mWDW and AASW, underneath the ice shelf ...

**3.24 Figure 5 caption – on the second line, remove "or" between "during" and "ebbing"** Done.

**3.25 L227:** What was the wind here? In general, I think a more systematic treatment of the winds is required. The values seem to be presented for the first time at random places in the Discussion, which makes it confusing. These values could perhaps be presented in the results?

We recognise that this is confusing. We have now included the minimum, maximum, and average wind speed in Table 1 in the methods section, where we first mention that the minimum, maximum, and mean wind speeds are used in the calculation of the CO2 flux. We refer to the table in lines 117-118 (of the original manuscript).

In the discussion section, we have now included a reference to Table 1 in line 227, in the caption of Figure 8, and in line 342. We kept the repetition of the range of wind speeds mentioned for each tidal observation period in lines 235 and 245.

We hope that the reader will find this more consistent and can more easily look up the wind speed values now that they are displayed in Table 1.

In doing so, we have slightly altered the caption of Table 1: Original text: Details of the tidal observations made with repeat CTD casts ... Revised text: Details of the tidal observations **based on** repeat CTD casts ...

We also corrected a typo that we found during revisions in the wind speed unit on line 246 of the original manuscript: Original text: (3 - 10 m-1) Corrected: (3 - 10 m s-1)

**3.26** L232: The idea of a "salinity front" is an important one that crops up throughout the Discussion. However, I don't think it is introduced and explained in a systematic way (unless I missed something). The authors should introduce this idea clearly early on in their Discussion so that its later significance is obvious to the reader.

We agree with this constructive input by the reviewer. Line 224 in the original manuscript is where we mention a salinity front for the first time (with reference to Skogseth et al., 2013). The next line is where we aimed to link it back to our polynya case study. We have adapted lines 229-232 in the original manuscript to give a better introduction.

**Original text lines 229-232:**

... In the latter study, the tidal variability was characterised by a cold salinity front that moved back and forth with the tide.

In our case study, the input of fresher water with lower DIC and TA content is likely advection of AASW from the north-east of the front (and sampling site), influenced by the summer sea ice melt.

**Revised text:**

... In the latter study, the tidal variability was characterised by a cold salinity front that moved back and forth with the tide. This "salinity front" is characterised by lower and higher salinities in the water column on either side of a sharp horizontal salinity gradient. A salinity front of this description might have been moving back and forth with the tide over the sampling site in the Ekström polynya, which from a Eulerian perspective resulted in the properties of the water column changing as shown in Fig. 3 and 4. In our case study, the fresher water with lower DIC and TA content on the northern side of the salinity front (and sampling site) is likely advection of AASW, influenced by summer sea ice melt.

Line 250 changed from "The other side of the front ..." to "The other side of the **salinity** front ..."

3.27 Figure 6: a) Is the lack of fluorescence variability during PS117 the result of the colour scale chosen? b) Also, why are some of the data missing from panel C? c) Finally, the fluorescence data in panel E seem to me to support the authors' arguments about tidal mixing but they're not discussed at all, nor integrated into the authors' arguments – I think they probably should be.

a) The fluorescence values measured during PS117 are similar to those measured in the deep ocean at CTD stations where the depth is >>2000 m, even though they vary at values < 0.1. We decided to put the PS117 data on the same scale as for PS89, because it highlights the contrasts in variability between them. Even when we plot the fluorescence for PS117 on its own scale, the variability over time is not very obvious. As the variability in the PS117 for fluorescence is very small and the number of PS117 profiles we have for this parameter is less than for the others (see comment 3.12), we think that changing the colour scale will not contribute enough to the messages of the manuscript. However, to clarify the range of variability in the PS117 data, we have added the following to the caption of Fig. 6:

**Added text to caption Figure 6: Fluorescence during PS117 varied at values < 0.09.**

b) We have a lower number of casts from which we have fluorescence data than for other parameters. Please see our response to comment 3.12, where we provide the explanation.

c) We agree with the reviewer that we can better incorporate the fluorescence results into the discussion to support our arguments. To do so, we have made the following changes, which include addition of text on lines 231-234, a minor change to the structure of the text by moving lines 256-258 down to line 266 and adding a few lines, and also making minor revisions to lines 310 and 320. Note that some of the revisions include those made in response to comment 3.26.

**Original text lines 231-234:**

In our case study, the input of fresher water with lower DIC and TA content is likely advection of AASW from the north-east of the front (and sampling site), influenced by the summer sea ice melt. In addition to a dilution effect, the accompanying increased fluorescence signal during rising tide on PS89 suggests that photosynthesis in this water has likely contributed to its lower DIC, TA, and nutrient content (Fig. 4, 6), which is sustained by solar radiation.

**Revised text lines 231-234:**

In our case study, the fresher water with lower DIC and TA content on the northern side of the salinity front (and sampling site) is likely advection of AASW, influenced by summer sea ice melt. Based on the fluorescence increase during rising tide for PS89 (Fig. 6E), the water on this side of the front seems to be richer in phytoplankton cells compared to the southern side of the front. In addition to a dilution effect, the accompanying increased fluorescence signal during rising tide suggests that photosynthesis in this water has likely contributed to its lower DIC, TA, and nutrient content (Fig. 4, 6), which is sustained by solar radiation.

Original text lines 256-258 moved down to line 266:

In terms of biogeochemical properties, the sub-ice shelf water is expected to be less ventilated compared to the AASW and to have relatively higher nutrient and DIC content, and lower dissolved oxygen content, as a result of net remineralisation (Fig. 4, 6).

Revised text lines 266-268:

In terms of biogeochemical properties, the sub-ice shelf water is expected to be less ventilated compared to the AASW and to have relatively higher nutrient and DIC content, and lower dissolved oxygen content, as a result of net remineralisation. Due to the lack of exposure to the atmosphere, phytoplankton cells (for which we use fluorescence as a proxy) are expected to be mostly absent. This description of water properties is consistent for the properties observed during ebbing tide (Fig. 4, 6). It therefore seems feasible that less ventilated, colder water from underneath the ice shelf with lower oxygen and higher nutrient and DIC content can extend to the edge of the ice shelf during ebbing tide and into the polynya.

Added text on line 310: This also applies to times at rising tide in the PS89 observation period when the fluorescence signal increases in the water column, suggesting advection of phytoplankton cells into the polynya.

Original text on line 320: Outflow and mixing of ice shelf melt water might have been stronger during PS117 than PS89.

Revised text on line 320: Earlier, we noted that southerly winds during PS117 may have counter-acted some of the advection of fresher, more ventilated water from the northeast during rising tide. However, the nutrient and oxygen content, as well as the very low fluorescence during PS117, suggest that the outflow and mixing of ice shelf melt water might have been stronger during PS117 than PS89, as well.

**3.28 L240: See above about "salinity front".**

Indeed, please see our response to comment 3.26. In addition, we have adapted line 243-244.

Original text lines 243-244: If a salinity front existed during PS117, it may have been located further away from the sampling site and closer to the sea ice edge of the polynya.

Revised text lines 243-244: If a salinity front existed during PS117, it could have been located further away from the ice shelf edge, and the sharp horizontal salinity gradient might therefore not have passed directly over the sampling site during PS117 as it did during PS89.

**3.29 L258: And presumably also due to a lack of exchange with the atmosphere?** Agreed! We have amended the text to include this point.

Original text: In terms of biogeochemical properties, the sub-ice shelf water is expected to be less ventilated

compared to the AASW and to have relatively higher nutrient and DIC content, and lower dissolved oxygen content, as a result of net remineralisation (Fig. 4, 6).

Revised text: In terms of biogeochemical properties, the sub-ice shelf water is expected to be less ventilated

compared to the AASW and to have relatively higher nutrient and DIC content, and lower dissolved oxygen content, as a result of net remineralisation **and lack of exchange with the atmosphere** (Fig. 4, 6).

**3.30 L260:** What do you mean here by "drill sites"? I think this is another example of an idea that needs to be better introduced.**

We agree with the reviewer that this idea can be better introduced. Smith et al. (2020a) measured the temperature and salinity profiles underneath the ice shelf by drilling through the ice shelf (with hot water) at various locations and lowering a CTD cast through it.

We have amended the text in a way that better introduces the work by Smith et al., before linking it to the relevance of our study. We have also added text to the caption of Figure 7 to improve clarity.

**Original text:** It is possible that the ebbing tide draws out water from underneath the ice shelf which is expected to be colder. Indeed, the study by Smith et al. (2020a) attributed variability in two repeat profiles at one of the drill sites (EIS-4) to tidal influences extending underneath the ice shelf.

Revised text: It is possible that the ebbing tide draws out water from underneath the ice shelf which is expected to be colder. Indeed, this possibility is supported by findings in Smith et al. (2020a), which includes a repeat profile of the Ekström Ice Shelf's cavity water at one of the measurement stations on the ice shelf (EIS-4; Fig. 7). These two repeat profiles were taken 11 hours apart. The difference observed in the vertical salinity and temperature profile between these two casts was attributed to tidal influences extending underneath the ice shelf (Smith et al., 2020a).

Original text caption Fig. 7: A) Temperature salinity diagram for PS89 (circles) and PS117 (diamonds) tidal observation periods, coloured according to dissolved oxygen concentrations, and the hot water drill CTD profiles through the ice shelf from Smith et al. (2020a) (coloured lines). Colours for hot water drill profiles correspond to the coloured marker locations on the map (B).

Revised text caption Fig. 7: A) Temperature-salinity diagram for PS89 (circles) and PS117 (diamonds) tidal observation periods, which are coloured according to dissolved oxygen concentrations. CTD profiles of the ice shelf's cavity water were collected and made available by Smith et al. (2020a) (coloured lines). The cavity CTD profiles were taken by hot water drilling through the ice at various locations on the ice shelf, which are shown on the map in (B) in corresponding colours to the profiles in (A).

**3.31** L281-282: **a**) Here again, it is not apparent to anyone not deeply familiar with the carbonate system why the nDIC and nTA data suggest that physical processes explain the observed variability. The rationale either needs to be explained here, or explained earlier (see my comment above) and then alluded to here. This is a pretty concept in support of the arguments made in the paper. **b**) Additionally, I got to the end of this paragraph as a whole (L293) and wasn't really sure what I should take away from it.

a) We have now included an explanation for the rationale in section 3.3 (see our response to comment 3.18). The sentence that the reviewer refers to here has been moved and modified to incorporate revisions in response to the second part of this comment (see below). As part of those revisions, we have inserted a reference to the explanation we newly provided in section 3.3.

b) We thank the reviewer for this comment. Upon re-reading the paragraph that the reviewer is referring to, we agree. We have moved this paragraph down, because we think it improves the flow of the discussion. We have placed it after discussing the influence of sub ice shelf water in the polynya. As we made a few other revisions to this section (in response to the first part of this comment, comment 3.32, and comment 2.2), we would like to refer the reviewer to pages 15 and 17 of the revised manuscript to see the result of the combined revisions made to this part of the text.

**Page 15 of the revised manuscript:**

As explained in Section 3.3, the salinity normalisation removes the impact of physical processes from the DIC and TA data. Therefore, any variability that remains in the nDIC and nTA results needs to be explained according to other processes. These processes are represented in Fig. 5 by theoretical lines that indicate how nTA and nDIC would change relative to each other as a result of photosynthesis/respiration, CaCO3 dissolution/precipitation, and CO2 uptake/release (Zeebe & von Gladrow, 2001). For example, factors that could be relevant to net photosynthesis are variable sea ice cover affecting light availability, nutrient replenishment during ebbing tide, and mixing of phytoplankton cells into deeper water during rising tide (Gleitz et al., 1994). Yet, none of these processes seem particularly dominant in changing the nTA and nDIC content (Fig. 5). The results in **Fig. 5** show a legacy of processes that may have occurred in the past weeks to months, as the **marine carbonate** system's equilibration time with the atmosphere is slow, especially in sea ice covered regions. Additionally, the data during rising tide might also reflect processes that happened in the sea ice, which will have affected the carbonate chemistry of the sea ice melt water and thus the properties of the AASW. While we here consider the tides to transport a salinity front back and forth across the sampling site, we must also recognise that the sampled mass of water on each side of the front is not exactly the same during each tidal phase, which contributes to the variability observed in the dataset.

**Page 17 of the revised manuscript:**

In this study, we have argued that the DIC variability in the coastal polynya is driven by back-and-forth movement of water under the force of tidal currents across the sampling site located in a region where there is a horizontal gradient in DIC content: lower DIC content to the north-east, influenced by summer sea ice melt, and higher DIC content to the south-west, influenced by unventilated ice shelf cavity water. This led us to investigate whether there is evidence in our dataset for a tidally-driven horizontal DIC pump. For example, net transport away from the ice shelf could transport DIC and nutrients (and perhaps even iron) from the ice shelf towards surface waters on the continental shelf that are exposed to sea ice and the atmosphere. Subsequent biological carbon uptake will then remove DIC. However, when we calculate the trajectory of a water parcel (using the ship's position as a starting point and the average current velocity of the water column) the net transport is south/south-east, i.e. towards the ice shelf (not shown). This implies that surface waters would be modifying the properties of the water underneath the ice shelf over time (instead of the other way around), for example by dilution. If this is the case, we would expect to see a trend in the DIC content of the polynya during ebbing tide. However, this is not the case and the net change in DIC

content over six hour periods (including ebbing and rising tide) is zero (Fig. G4 for PS89). Our observations are a snapshot of a highly dynamic system and consequently they do not provide enough data to analyse such modifications of the seawater physico-chemical properties. Nevertheless, they can be the beginning of future studies into this topic.

**3.32 L297: Please explain the significance of the nTA:nDIC ratio.**

By referring to the nTA:nDIC ratio, we aimed to refer to how nTA relates to nDIC in Fig. 5. However, we now understand that this may have been confusing because we only use the term "nTA:nDIC ratio" once and have not given specific introduction to the theoretical lines in Fig. 5, which represent processes that would change the relationship of nTA to nDIC along these lines. This is a comment that relates to comment 2.2 (by reviewer #2), which suggested to give an introduction to these processes. We think that our minor edit to line 297, combined with the changes we made in response to comment 2.2, improve the clarity.

Original text line 297: Yet, none of these processes seem particularly dominant or persistent in the nTA:nDIC ratio (Fig. 5).

Revised text: Yet, none of these processes seem particularly dominant in changing the nTA and nDIC content (Fig. 5).

**3.33 L299: The word "data" is plural.**

"This data ..." has been corrected to "These data ...".

**3.34 L313: What do you mean by "light stability" in this context?**

We recognise that the original phrasing is unclear. We meant that sustained primary productivity needs enough light availability in the upper part of the water column that does not vary too much on a short time scale. However, as this is driven by the MLD (affecting the depth of phytoplankton cells) and we are referring in the text to drivers of *low* primary productivity (and not drivers of high primary productivity), we have removed "light stability" in the sentence:

Original text: ... and the most likely drivers of low primary productivity are increased MLD, light stability, and grazing pressure (Arrigo and van Dijken, 2003).

Revised text: ... and **important** drivers of low primary productivity are increased **MLD** and grazing pressure (Arrigo and van Dijken, 2003).

**3.35 L334: I think this section might benefit from a sentence at the end here summing up what we have learned from the new data presented here rather than simply ending the section after outlining everything we don't know.**

We like this suggestion! We have added the following at the end of this section:

**Added text on line 334 of the original manuscript:**

Despite the unknowns outlined above, the case studies presented in this work show that strong tidal influences on the physical structure and biogeochemical properties of the water column can be expected along the Weddell Sea coastline (and other polar regions subject to strong tides), especially in close proximity to ice shelves and regions of sea ice melt. They also show that local winds and ice shelf meltwater outflows can increase the complexity of the tidal impact within in a region such as a coastal polynya. In addition to studies on the physical role of tides on (for example) basal ice shelf melt, ecological, biogeochemical, and air-sea gas exchange studies can benefit from a better understanding of tidal impacts on the water column.

3.36 L345: "...fluctuates between the bottom (incoming tide) and... (outgoing tide)" – the meaning of this sentence is unclear. We have improved the sentence, as shown below:

Original text: However, the depth at which the fCO2 of the seawater is equal to that of the atmosphere (marked by a dashed line in Fig. 4A) fluctuates between the bottom (incoming tide) and the surface (outgoing tide).

Revised text: However, the depth at which the fCO2 of the seawater is equal to that of the atmosphere (marked by a dashed line in Fig. 4A) fluctuates **from near the bottom of the water column during incoming tide to the surface during outgoing tide.**

**3.37 L351: In general, I think it's better to avoid hyperbolic terms such as "drastic".** We have replaced "even more drastic" with "even stronger".

Original text: ... an even more drastic fluctuation ... Revised text: ... an even **stronger** fluctuation ...

3.38 Figure 7 – a) I believe that the use of PSU for practical salinity and outdated and such data should be presented with units. Alternately, absolute salinity should be used.
b) Additionally, what is AB? c) And finally in the 5th line of the caption, should the reference to "single purple markers" actually be to "single whitecrosses"?
a) We have now updated this in Fig. 7.

b) AB stands for Atka Bay, where Smith et al. (2020a) also took some profiles. We have now incorporated an explanation in the caption:

Added text in caption of Figure 7: B) Map of measurement locations of the Ekström Ice Shelf cavity CTD profiles by Smith et al. (2020a), denoted by "EIS\_", a measurement location in Atka Bay by Smith et al. (2020a) denoted with "AB", and the sampling location of the tidal observations indicated by the yellow star.

c) Indeed. "Single purple markers" have been changed to "**single white crosses**" in the caption of Figure 7.

**3.39 L378:** There's some odd repetition here that makes the sentence confusing – please revisit for clarity.

We thank the reviewer for noticing this. We have re-written these lines to remove the odd repetition and to improve the clarity.

Original text: We emphasise the potential misrepresentation of the role of coastal polynyas in the Weddell Sea CO2 uptake if tidal influences are not accounted for using two extreme scenarios from the hypothetical case where samples are collected only during rising or ebbing tide of the PS89 tidal observation: samples are only taken at either peak rising tide or peak ebbing tide, which lead to an overestimation or underestimation of the CO2 flux, respectively (see above).

Revised text: We emphasise the potential misrepresentation of the role of coastal polynyas in the Weddell Sea  $CO_2$  uptake if tidal influences are not accounted for. For this, we again use the two extreme scenarios based on the PS89 observations that were also used above to illustrate the maximum potential over- and underestimation of the  $CO_2$  uptake. I.e., we use the hypothetical cases where seawater samples are either collected at peak rising tide (overestimation of  $CO_2$  uptake) or at peak ebbing tide (underestimation of  $CO_2$  uptake).

**3.40** Figure 8 – where is the "dotted line" referred to in the 3rd line of the caption? And if the filled shading indicates the range, does the black line indicate the average wind speed?**

The line with the information on the "dotted line" was accidentally left in the caption, but should have been removed. The parameterisation by Sweeney et al. (2007) was used in the  $CO_2$  calculations by Brown et al. (2015). Since we are making a comparison to their work, we wanted to use the same parameterisation. However, the results show no clear difference and we have therefore omitted it from the plot. We have now removed this line from the caption.

The full black, green, and orange lines indeed indicate the fluxes based on the average wind speeds. We have now included this information in the caption.

Original caption: Air-sea CO2 flux (in mmol m-2 day-1 on the left y-axis and in mol m-2 year-1 on the right y-axis) determined from the discrete surface seawater sample measurements for the PS89 (green) and PS117 (orange) tidal observation periods, and from the PS89 underway fCO2 measurements (black), which started on the 7th of January 2015. Dotted lines use the parameterisation by Sweeney et al. (2007), which was used in calculations by Brown et al. (2015). The filled shading indicates the range of the flux calculated using the minimum and maximum wind speed measured during PS89 and PS117, respectively. Negative flux represents CO2 uptake by the ocean.

Revised caption: Air-sea CO2 flux (in mmol m-2 day-1 on the left y-axis and in mol m-2 year-1 on the right y-axis) determined from the discrete surface seawater sample measurements **and average wind speed** for the PS89 (green) and PS117 (orange) tidal observation periods, and from the PS89 underway fCO2 measurements (black), which started on the 7th of January 2015. The filled shading indicates the range of the flux calculated using the minimum and maximum wind speed measured during PS89 and PS117, respectively (**Table 1**). Negative flux represents CO2 uptake by the ocean.

**3.41 L414: What do you mean by "mediating effects"?**

We recognise that this sentence is unclear. We also realised that "mediating" is the wrong word for what we meant to express. A better word is "modulating". The line has been revised, as shown below:

Original text: The datasets here are too small to explore the mediating effects of these processes.

Revised text: The datasets of the two short case studies presented here are too small to fully explore the modulating effects of these processes on the water column variability.

**3.42** L415: What is meant by "repeats are required"? Also, I found the inclusion of "carbonate chemistry" a little incongruous here since that is what is presented in this manuscript. Can you clarify your meaning?**

We meant that case studies such as these tidal observations would have to be repeated to verify and study interacting processes. "Carbonate chemistry" was included in the list, because we meant to convey that an array of measurements and samples, in addition to carbonate chemistry measurements, are required to constrain relevant interacting processes. We have modified the text as shown below to improve the clarity.

Note: the modified text follows the text edited according to comment #3.41 (see above). Also note: we have inserted a break in the paragraph after the modified text, because we think it improves the readability of the text.

Original text: Longer tidal observations and repeats are required, along with measurements of micro-nutrients, carbonate chemistry, biological productivity, and oxygen isotopes, to be able to constrain interacting processes. A better understanding of the carbonate chemistry of the water underneath the ice shelf - although challenging to obtain - would help understand the influence of this water during ebbing tide.

Revised text: **To be able to do so,** longer tidal observations are required **that cover different parts of the spring-neap tidal cycle, and at different times of the year to capture varying wind and ice melt/growth conditions.** Alongside carbon system state variables, an array of co-collected measurements, such as micro-nutrients, biological productivity, and oxygen **stable** isotopes, **can help** to constrain interacting processes. **An** understanding of the carbonate chemistry of the **cavity** water underneath the ice shelf - although challenging to obtain - would help understand the influence of this water **on the polynya** during ebbing tide.

**3.43 L424: Please see my comments above about the "salinity front".**

Please see our response to comment 3.26. Additionally, we have revised line 423-424:

Original text lines 423-424: It may also help identify the formation and characteristics of - what is described here as - a salinity "front" that moves back and forth with the tide.

Revised text lines 423-424: It may also help identify the formation and characteristics of **a horizontal coastal salinity gradient – here referred to as a "salinity front"** – that moves back and forth with the tide.

**3.44 Data availability: is it acceptable to the journal for the data to only be accessible by contacting the corresponding author? This seems unusual.**

We believe there might have been a slight misunderstanding caused by the manner in which we had originally written our statement for the data availability and would like to clarify. As stated in our data availability statement, our PS117 data will be made available online on the Pangaea database. We were still working on this at the time of initial submission of the manuscript (also coordinating how the rest of the data of PS117 will be published). The data used in the current work has since been submitted to Pangaea and we have received the preliminary citation for it, which should be finalised before finalisation of the manuscript. The editors are aware of this.

Original text: For data access, contact the corresponding author. DIC and TA datasets will be made available online on the Pangaea database (in progress).

**Revised text: Data are available on Pangaea: https://doi.org/10.1594/PANGAEA.946363.**

**3.45 Figure A1-A4 – what does the pink star denote? This information should be included in the caption.**

The pink start denotes the sampling location within the polynya during the PS89 and PS117 tidal observations. We have now included this in the captions for Figures A1-A4.

3.46 Figure G4 – I think the authors should offer a reason for the anomalous silicate data from PS117. Are they certain it's not an analytical error? If not, might it shed some light on the water mass encountered during PS117? I found it a little odd that all these data were presented, the inconsistency highlighted, and then no discussion/explanation offered. Same comment applies to Figure G6. The silicate data from PS117 are not anomalous. There are different relationships between salinity and all three nutrients for PS89 and PS117 (i.e. not only for silicate). Obviously, the water masses present in 2015 (PS89) and in 2019 (PS117) had different characteristics. This is shown in G4, G5, and G6. We have not found any measurement errors, and based on the data also have no reason to presume that.

It is indeed expected that nitrate and phosphate will be more similar to each other because the processes in which they are involved are similar (i.e., utilization and remineralization of organic matter). This is different for silicate, as it is incorporated in opal. Opal is formed and dissolved independent on the organic matter processes in which nitrate and phosphate are changed. We can therefore expect different relationships for PO4/NO3 on the one side and silicate on the other; and these relationships may be different in different years.

Please see our response to comment 3.20, part b), in which we explain the changes made to the manuscript. After incorporating the revisions, we decided to remove figure G6, as the reviewer's comments made us realise that it causes more confusion than clarification. We feel that it no longer adds any important information that Figs. G4 and G5 do not already provide and would support our argument, which we have now clarified (as shown in response to comment 3.20b).